# SEC-MX: an approach to systematically study the interplay between protein assembly states and phosphorylation

Ella Doron-Mandel [1,4] ✉, Benjamin J. Bokor[1,4], Yanzhe Ma[1,4], Lena A. Street[1], Lauren C. Tang[1], Ahmed A. Abdou[1], Neel H. Shah [2], George Rosenberger [3] & Marko Jovanovic [1] ✉

A protein's molecular interactions and post-translational modifications (PTMs), such as phosphorylation, can be co-dependent and reciprocally co-regulate each other. Although this interplay is central for many biological processes, a systematic method to simultaneously study assembly states and PTMs from the same sample is critically missing. Here, we introduce SEC-MX (Size Exclusion Chromatography fractions MultipleXed), a global quantitative method combining Size Exclusion Chromatography and PTM-enrichment for simultaneous characterization of PTMs and assembly states. SEC-MX enhances throughput, allows phosphopeptide enrichment, and facilitates quantitative differential comparisons between biological conditions. Conducting SEC-MX on HEK293 and HCT116 cells, we generate a proof-of-concept dataset, mapping thousands of phosphopeptides and their assembly states. Our analysis reveals intricate relationships between phosphorylation events and assembly states and generates testable hypotheses for follow-up studies. Overall, we establish SEC-MX as a valuable tool for exploring protein functions and regulation beyond abundance changes.

A major effort in studying the dynamics of biological systems is to measure differential changes in the proteome. Classically, proteome dynamics have been studied by measuring differences in total protein expression levels. However, total proteome levels do not convey the full array of post-translational dynamics. For example, the protein's assembly state – whether a protein is acting alone as a monomer or as part of different complexes – often changes between biological conditions and systems, despite similarities in overall expression levels[1–3]. These differences in assembly states, mainly driven by protein-protein interactions (PPIs), may reflect distinct functions that contribute to differential cell states.

An additional central mechanism contributing to the complexity of the proteome is the attachment of functional groups to proteins, referred to as post-translational modifications (PTMs)[4]. Notable examples encompass ubiquitination, acetylation, methylation, and phosphorylation – which is one of the most extensively studied. PTMs can induce various alterations in protein activity, including activation or inhibition, tagging for degradation, and subcellular localization, among others. Notably, most PTMs and PPIs are dynamic, playing roles in regulatory mechanisms and changing in response to different biological conditions[5–9]. Therefore, characterizing these dynamic states is essential for a comprehensive biological understanding.

Cumulative research over several decades has demonstrated the interdependence of a protein's assembly state and PTM-status[5,8–11]. For instance, phosphorylation can modulate a protein's interaction interface, and conversely, protein interactors can obstruct phosphorylation sites, thereby denying access to kinases or phosphatases. Past works on the interplay between PPIs and PTMs have relied on targeted

[1]Department of Biological Sciences, Columbia University, New York, NY, USA. [2]Department of Chemistry, Columbia University, New York, NY, USA. [3]Department of Systems Biology, Columbia University, New York, NY, USA. [4]These authors contributed equally: Ella Doron-Mandel, Benjamin J. Bokor, Yanzhe Ma. ✉e-mail: ed2853@columbia.edu; mj2794@columbia.edu

methods, such as co-immunopurification, to enrich the interactomes of specific proteoforms[5,8]. However, these approaches are limited in their scope and a method to simultaneously study assembly states and PTMs on a global scale is critically missing.

Previously, Size Exclusion Chromatography followed by Mass Spectrometry (SEC-MS) has been used to analyze assembly states, identify PPIs and characterize the composition of molecular-complexes[1,12–26]. SEC is used to separate different assembly states of the same protein, which elute in distinct fractions based on the molecular weight (MW) of each assembly (for example, a monomer will elute in a low MW fraction versus a multimeric complex of a higher MW[2]). However, SEC fractions are limited by input amounts that render PTM enrichment for SEC-MS challenging. As a result, previous works have mined phosphopeptides in SEC datasets by meta-analysis[27], or studied the effects of phosphatase-treatment on SEC elution profiles[28]. However, no study to date has directly measured enriched phosphopeptides with SEC-MS.

Here, we present a global, quantitative methodology that enables the simultaneous characterization of PTMs and assembly states within the same sample, allowing us to explore their relationship across biological conditions. Expanding on prior SEC-MS and co-fractionation multiplexing works[29], we use isobaric tags to develop Size Exclusion Chromatography fractions MultipleXed (SEC-MX) and demonstrate its advantages in: (1) reducing the number of LC-MS/MS runs required to reconstruct the PPI network, (2) enabling phosphopeptide enrichment and measurements thereby allowing characterization of PTMs along the SEC range, and (3) simplifying quantitative comparisons between biological conditions that are multiplexed together.

In this study, we employed SEC-MX to comprehensively characterize the SEC elution profiles of both non-modified and phosphorylated peptides from HEK293 and HCT116 cells – two human cell lines that differ in the steady-state activation of signaling-pathways due to their distinct driver modifications. HCT116 is of cancer origin[30], while HEK293 are immortalized via adenovirus transformation[31]. This yielded a dataset of concurrent non-targeted measurements of phosphorylation events and assembly states for thousands of proteins across two distinct biological conditions. Our analysis approach then enabled a comparative examination of assembly states and their phosphorylation status between the two cell lines. Overall, our study provides insights into the intricate interplay between post-translational modifications and protein assembly states, under-scoring the value of SEC-MX in unraveling the complexities of protein regulation.

## Results

### Development and benchmarking of SEC-MX

We developed SEC-MX (SEC fractions MultipleXed) to enable quantitative comparisons between different samples and to measure phosphorylation events on proteins in distinct assembly states by enriching phosphopeptides from SEC fractions. First, we set out to multiplex SEC fractions to increase the sample yield for phosphopeptide enrichment. Multiplexing was achieved by labeling SEC fractions with isobaric tags. Tandem mass tags (TMTpro) were used because they provided the highest number of labeling channels available, allowing combinations of up to 18 samples in a single liquid-chromatography tandem mass spectrometry (LC-MS/MS) run. We multiplexed SEC-adjacent fractions within the same TMT mix to minimize the occurrence of missing data points and reduce sample complexity (Fig. 1A, B, Supplementary Fig. 1A). In doing so, we leveraged the fact that, once a peptide is triggered for MS2 acquisition, TMT reporter intensity values are typically assigned to all channels, resulting in more complete elution profiles. In addition, we designed a "full-overlap" scheme (Fig. 1B, Supplementary Fig. 1C) in which every fraction is measured twice, in different mixes, increasing coverage and enabling batch correction

between different TMT-mixes (see Methods section, Supplementary Fig. 1).

We compared SEC-MX to the field's current gold standard – label-free SEC followed by data independent acquisition (DIA) for protein identification and quantification (SEC-DIA). To this end, we conducted SEC-MX and SEC-DIA on HEK293 cells in duplicate biological replicates. We found that coverage in SEC-MX was comparable to SEC-DIA, with only a 10% difference in protein-group identifications, despite a 7-fold reduction in the number of LC-MS/MS runs (57 versus 8 in DIA or TMT, respectively, Fig. 1C). Additionally, SEC-MX produced SEC elution patterns very similar to those of SEC-DIA, as demonstrated by the comparable intensity heatmaps and the high correlation between elution profiles measured by both methods (Fig. 1D–F).

We then used the network-centric analysis algorithm SECAT[22] to identify high-confidence PPIs. We observed that SEC-MX identified a similar number of PPIs as did SEC-DIA, (3,588 and 4,208 at 5% false-discovery rate, respectively, of which ~54% were overlapping), with similar network parameters (Fig. 1G, H, Supplementary Fig. 2A–C). The data was also analyzed with a reference-free PPI analysis tool to identify previously-uncharacterized interactions (EPIC[24]), resulting in 12,620 and 14,406 PPIs in SEC-MX and SEC-DIA, respectively (Supplementary Fig. 2D–F). Together, our observations showed that SEC-MX performs comparably to the field's gold standards in building a context-specific human PPI network while reducing measurement time.

### SEC-MX enables phosphopeptide enrichment

After validating the performance of SEC-MX for studying protein interactions, we next set out to test the feasibility of the multiplexing approach in enabling phosphopeptide enrichment using standard immobilized metal affinity chromatography (IMAC[32]). As mentioned above, we developed the method to facilitate differential analysis between biological conditions. Therefore, we designed the study to compare between two distinct cell lines, HEK293 and HCT116, as will be elaborated below. Fifty-four SEC fractions were collected from each cell line, labeled and multiplexed in the same TMT mixes to minimize variability in protein and phosphopeptide coverage between the two cell lines (Supplementary Fig. 1B). After labeling and multiplexing, 20% of each mix were taken for measuring the 'global' (unenriched) proteome for analysis of protein assembly states, while the rest of the sample was further allocated for phosphopeptide enrichment using IMAC (Fig. 2A).

Using this pipeline, we generated two datasets: the global dataset (gSEC) and phosphopeptide SEC dataset (phSEC), in 2 biological replicates each. We then conducted a union-based averaging across the replicates to reduce noise and allow more comprehensive comparisons between gSEC and phSEC (Supplementary Data 1 and 2, respectively). Furthermore, since different phosphopeptides could belong to distinct proteoforms they cannot be directly collapsed to the protein level. Therefore, we report all phSEC data at the peptide level and link it to the protein-level data of the corresponding parent protein in the gSEC dataset, rather than making comparisons strictly within the peptide or protein levels. Overall, the gSEC dataset yielded ~59,600 peptides per cell line, covering the same 5503 protein groups across both cell lines. The phSEC dataset recovered ~4600 phosphorylated peptides (with averaged enrichment efficiency of 96.6%, Supplementary Fig. 3A), spanning ~2150 proteins in each cell line (see Table 1, and Supplementary Table 1). Matching these datasets recovered 3827 phosphopeptides in phSEC that were linked to 1596 overlapping proteins in gSEC (Fig. 2B).

To assess the quality of the phSEC data, we compared the elution patterns of phosphopeptides in phSEC to their corresponding peptides in gSEC – observing a high degree of correlation (Fig. 2C, D, Supplementary Fig. 3B–D). For example, we examined the elution profiles of phosphopeptides associated with subunits of the stable

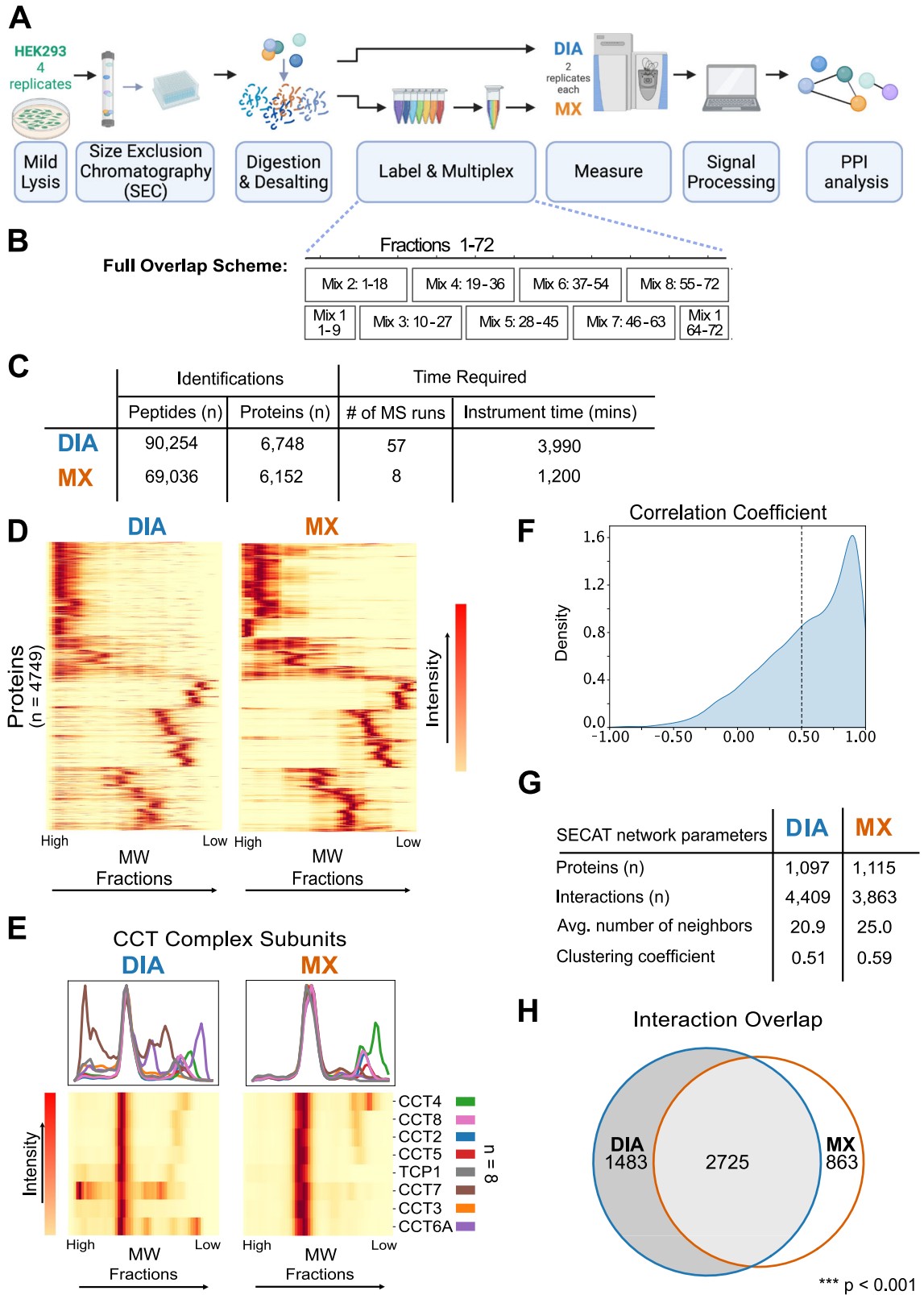

**F** Correlation Coefficient

**G**

| SECAT network parameters | DIA | MX |
|---|---|---|
| Proteins (n) | 1,097 | 1,115 |
| Interactions (n) | 4,409 | 3,863 |
| Avg. number of neighbors | 20.9 | 25.0 |
| Clustering coefficient | 0.51 | 0.59 |

**C**

| | Identifications | | Time Required | |
|---|---|---|---|---|
| | Peptides (n) | Proteins (n) | # of MS runs | Instrument time (mins) |
| **DIA** | 90,254 | 6,748 | 57 | 3,990 |
| **MX** | 69,036 | 6,152 | 8 | 1,200 |

**E** CCT Complex Subunits

CCT4, CCT8, CCT2, CCT5, TCP1, CCT7, CCT3, CCT6A (n = 8)

**H** Interaction Overlap

DIA 1483 — 2725 — MX 863

\*\*\* p < 0.001

CCT complex and found that they co-eluted with the fully assembled form of the complex (Fig. 2E, Supplementary Fig. 3E). In conclusion, our PTM enrichment downstream from SEC fractionation created a dataset measuring SEC elution profiles for ~4600 phosphopeptides and their matching parent proteins, which enables overlaying PTM-status with protein assembly states.

**A protein-centric analysis framework to study assembly states**

Next, we aimed to analyze our dataset in order to: (1) define assembly states for every measured protein, (2) map phosphorylation events onto assembly states, and (3) study how these patterns differ between conditions. Previously, SEC-MS has been widely used to identify pair-wise PPIs, characterize the composition of molecular-complexes, or

**Fig. 1 | Development and benchmarking of SEC-MX. A** Overview of experimental pipeline used to establish SEC-MX and compare it to SEC-DIA. PPI stands for Protein-Protein Interactions. Created in BioRender. Jovanovic, M. (2024). [https://BioRender.com/ s07v514][77]. **B** A full overlap mixing scheme was developed, in which each fraction is divided into two parts, and each half is measured in a different mix, keeping adjacent fractions together. Full details of the labeling and pooling scheme can be found in Supplementary Fig. 1. Created in BioRender. Jovanovic, M. (2024). [https://BioRender.com/ s07v514][77]. **C** Table showing protein and peptide coverage and measurement time in SEC-DIA and SEC-MX. **D** Heatmap representation comparing signals in SEC-DIA and SEC-MX, averaged across replicates for 4749 proteins measured in both (each row represents a protein). Columns represent fractions. Signal is row-normalized from 0 to 1 so that the max elution

peak per protein is represented in red. Rows in both heatmaps are arranged in the same order. **E** Elution traces and heatmap representation (subset from Fig. 1D) of the SEC elution of the CCT complex subunits ($n = 8$) in either SEC-DIA or SEC-MX. Traces are color-coded by subunit, as indicated to the right of the heatmaps. **F** Distribution of Pearson correlation coefficients between elution profiles in SEC-MX versus SEC-DIA per protein measured in both. 2947/4749 proteins have a correlation coefficient >0.5. **G** Parameters of the SECAT interaction networks based on SEC-DIA and SEC-MX data. **H** The overlap of interactions between SEC-DIA and SEC-MX, $q$-value < 0.05 in at least one condition and <0.1 in the other (see Methods section for details). The overlap is statistically significant (hypergeometric survival function, $\log_{10}$ ($p$Value) = -14,100, calculated over the background of all possible pairwise interactions of the proteins in the PPI network).

compare interactions between samples[1,18,22,24,33]. As opposed to a focus on interactions, recent work has shown the potential of a protein-centric approach that looks at the assembly state changes of individual proteins between conditions[2]. Based on this approach, we first identified assembly states by taking into account that each SEC elution peak represents a distinct assembly state of the protein (Fig. 3A). To this end we used a peak-calling algorithm (as detailed in the Methods section) to identify assembly states for each protein in each dataset independently (gSEC and phSEC for each cell line). This analysis identified that the ~5500 proteins in each gSEC sample are eluting in ~10,000 assembly states, with an average multiplicity of 1.9 peaks per protein (Table 1, Fig. 3B, Supplementary Fig. 3F), suggesting that over half the proteins in the cell are present in at least two distinct assembly states.

Following peak calling, each assembly state was categorized as monomeric or complexed based on the peak position along the SEC dimension (detailed in the Methods section, Fig. 3A). We observed that only ~18% of proteins eluted exclusively as monomeric, ~53% exclusively in their complexed form, and ~29% in both (Fig. 3C), suggesting that the majority of proteins in the cell are complexed with other molecules, in line with previous observations[1,2,15]. This trend toward assembled proteins was even more pronounced for phSEC profiles, where ~8% of proteins eluted exclusively as monomeric, ~82% exclusively in their complexed form, and ~10% in both (Fig. 3D). Comparing these distributions, we noted that the phSEC assembly states eluted less frequently as monomers than in gSEC. This discrepancy between the distributions of assembly states in gSEC and phSEC also holds true when analyzing the subset of proteins detected in both datasets (Supplementary Fig. 3G). Therefore, the lower percentage of monomeric assembly states in phSEC compared to gSEC suggests that phosphorylation occurs more often on the assembled form of a given protein, rather than its monomeric state.

In conclusion, we used a peak-focused approach to define assembly states from SEC data. Using this method, we demonstrated that more than half of the measured proteins were present in two or more assembly states, highlighting the importance of considering how alternative assembly states might be regulated. Therefore, we next turned to assign phosphorylation-events to specific assembly states.

**Mapping PTM onto assembly states**

To map phosphorylation onto individual assembly states, we aligned the peaks identified in phSEC to the peaks of their parent-protein in gSEC based on peak-apex position along the SEC range (Fig. 4A, see Methods). This analysis showed that ~94% of the phSEC proteins had at least one peak aligned between gSEC and phSEC, showing high agreement between the two datasets (Fig. 4B). When a gSEC peak aligned with a phSEC peak, we considered that as evidence that the protein is phosphorylated in the specific assembly state. Notably, 43% of the gSEC peaks for phosphorylated proteins were not aligned with a phSEC peak. This suggests that a substantial fraction of the phosphorylated proteins were selectively phosphorylated on specific assembly states. We next discuss a few of these examples.

For example, we found that non-muscle myosin IIA Myosin-9 (MYH9), an abundant actin-motor expressed in most eukaryotic cells, eluted in both a high MW peak (multimeric assembly state), and a smaller MW peak in the monomeric range (Fig. 4C, Supplementary Fig. 4A). The MYH9 phosphopeptides (spanning Serine-1943) co-eluted exclusively with the multimeric assembly state, suggesting differential phosphorylation of MYH9's two assembly states (assembled and monomeric). This observation is in-line with previous works showing that association of MYH9 with its binding partners is regulated by Serine-1943 phosphorylation[34,35].

In addition to multimeric assembly states determined by PPIs, assembly states may form by interaction with various other molecules, such as nucleic acids. For instance, Polypyrimidine tract-binding protein 1 (PTBP1), an RNA binding protein (RBP), eluted in two prominent assembly states: one high MW and another low MW. Interestingly, PTBP1 phosphopeptides eluted within a single peak overlapping the smaller MW assembly state (Fig. 4D, Supplementary Fig. 4B), suggesting that only the low MW assembly state of PTBP1 is phosphorylated. Being an RBP, one potential explanation to this phenomenon could be that phosphorylation regulates the association of PTBP1 to RNA transcripts. To test this hypothesis, we plotted the elution profile of PTBP1 in SEC from HEK293 cells, before and after RNAse digestion (Fig. 4E). We observed that the intensity of the high MW peak of PTBP1 is decreased following RNAse digestion, and the smaller MW peak is increased – supporting the notion that the phosphorylated PTBP1 form is no longer interacting with RNA. A similar pattern was observed in our data for various other RBPs including CPSF3, which like PTBP1 seems to be phosphorylated only in its RNA-free form (Supplementary Fig. 4C). On the other hand, other RBPs, like DDX54 (Supplementary Fig. 4D), exhibited the opposite pattern with phSEC peaks only matching the high MW RNA-bound assembly state. Together, these observations suggest a role for phosphorylation in regulating RBP interactions with RNA. Furthermore, this example highlights the advantage of using a protein-centric approach to analyze SEC data, as opposed to a PPI-focused approach – namely, the ability to identify assembly state changes without the requirement to identify a protein's underlying interactors.

Lastly, we found that many proteins had multiple phosphopeptides – associated with different phosphosites. In some cases, such as with DNA replication licensing factor MCM3 (MCM3), these distinct phosphosites were associated with different assembly states. MCM3 is a subunit of the minichromosome maintenance 2–7 complex (MCM2-7), a hetero-hexameric complex that functions as a DNA replication licensing factor. It is loaded onto replication origins to form inactive pre-replicative complexes, which are then activated by kinases to form the active CDC45-GINS-MCM helicase complex[36,37]. In gSEC, MCM3 eluted in two assembly states: the full hexameric complex (higher MW) and a lower MW complex with MCM5 and the auxiliary protein MCM binding-protein (MCMBP), (Fig. 4F, G, Supplementary Fig. 4E, F). In phSEC, we identified three distinct phosphopeptides of MCM3, each containing known phosphorylation sites; one spanning amino acids (AA) 668-689 including Serine-672 and Serine-681, the other spanning

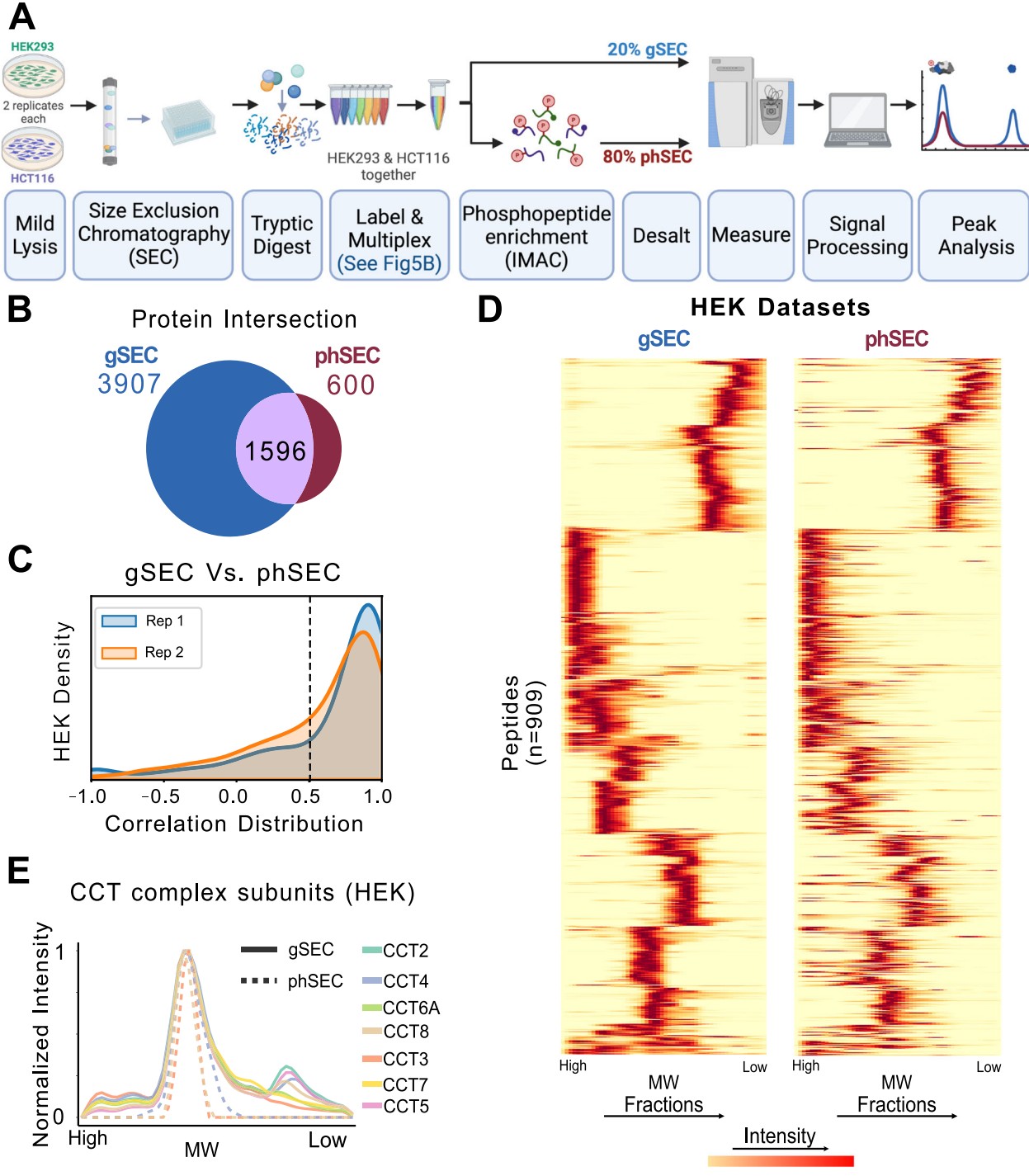

**Fig. 2 | SEC-MX enables phosphopeptide enrichment. A** Overview of the experimental workflow of SEC-MX. The cells were lysed, fractionated by size-exclusion chromatography, digested, and labeled. After pooling, 80% of each 18-fractions mix were allocated for phosphopeptide enrichment and measurement of phSEC, while the remaining 20% were processed for gSEC. Created in BioRender. Jovanovic, M. (2024). [https://BioRender.com/ s07v514][77]. **B** Venn diagram showing the overlap of protein identifications between the gSEC and phSEC datasets. **C** Distribution of Pearson correlation coefficients between gSEC and phSEC elution profiles for each overlapping peptide in the HEK293 dataset, per replicate (replicate 1 in blue, replicate 2 in orange, as indicated in the key). 456 / 627 and 343 / 522 peptides have correlation coefficients greater than 0.5, in replicate 1, and 2,

respectively. **D** Heatmap representation of the elution profiles of peptides overlapping between gSEC and phSEC (HEK293 data). The peptide number ($n = 909$) represents the number of rows and is based on the intersection of peptide identifications across the datasets (gSEC/phSEC), in the HEK293 signal averaged between replicates. Columns represent fractions. Signal is scaled so that the max elution peak per protein is represented in red. Rows in both heatmaps are arranged in the same order. **E** Elution traces for CCT complex members identified in gSEC and phSEC from the average of two replicates in HEK293 cells. Traces are color-coded by subunit, as indicated to the right of the plot. (Similar results were obtained for HCT116 cells, shown in Supplementary Fig. 3E).

AA 701–724 including Tyrosine-708 and Threonine-722, and the third spanning AA 725-732 including Serine-728. Interestingly, while phPEP[725-732] co-eluted with both gSEC peaks, phPEP[701-724] and phPEP[668-689] eluted only in the lower MW peak (Fig. 4F, Supplementary Fig. 4E). Of note, a recent study suggested that MCMBP may play a role in forming the MCM2-7 hexameric complex before it is loaded onto the chromatin[38], meaning that the MCM3-MCM5-MCMBP complex is potentially a stable intermediate assembly product of the full complex. Therefore, the observation that phPEP[668-689] and phPEP[701-724] elute only with the MCM3-MCM5-MCMBP assembly state raises the hypothesis that the sites within these peptides are dephosphorylated prior to the assembly of the full complex. While further studies are required to test this hypothesis, these observations exemplify the potential of our method in formulating testable hypotheses of how phosphorylation contributes to individual assembly states and their functions.

In conclusion, we used a peak-focused analysis to map phosphorylation events to individual assembly states. We showed how this approach can be used to explore the relationship between protein-interactions and PTM regulation. For example, by identifying differential phosphorylation between assembly states, such as in cases where the monomeric and assembled forms differ in phosphorylation (MYH9), how phosphorylation may regulate the binding of proteins to nucleic acids (PTBP1, CPSF3, DDX54), or how distinct phosphorylation-sites are associated with different assembly states (MCM3). Next, we expanded the analysis to compare assembly states and their phosphorylation between the two analyzed cell lines.

## SEC-MX enables differential comparison between biological samples

One of the main considerations in designing SEC-MX was to facilitate differential analyses of assembly state changes between biological samples and/or conditions. Many studies mapped the human protein-interaction network from HEK293 cells[15,20,22,29,39–43], including a recent seminal work that contrasted the interactome of HEK293 cells with that of HCT116 cells[44] based on affinity-purification mass-spectrometry (AP-MS, hereafter referred to as "BioPlex"). The BioPlex network covered over 10,100 human proteins measured in both HEK293 and HCT116, showing that approximately 50% of protein interactions are shared between these cell lines. Additionally, the BioPlex network revealed that the interactions exclusive to each cell line's network reflect specialized biology related to their distinct tissues of origin and driver modifications. HEK293 cells were cultured from a female embryo kidney, immortalized by adenovirus transformation[31]; whereas, HCT116 cells are derived from an adult male colorectal carcinoma[30], carrying a KRAS mutation[45] (Fig. 5A). We reasoned that contrasting the same two cell lines as BioPlex (HEK293 and HCT116) would provide SEC-MX a point of reference. Furthermore, by using the same system, SEC-MX offers the field a complementary resource to

**Table 1 | Identifications per cell-line across both replicates**

| Condition | Dataset | Peptides | Proteins | Peptide AS | Protein AS |
|---|---|---|---|---|---|
| HCT116 | gSEC | 59,658 | 5503 | 88,988 | 10,501 |
| | phSEC | 4674 | 2170 | 6743 | (5044) |
| HEK293 | gSEC | 59,615 | 5503 | 86,007 | 10,281 |
| | phSEC | 4602 | 2141 | 6129 | (4612) |
| Union/Aligned | | 63,402 | 6103 | 98,338 | 11,974 |

(AS = Assembly States).

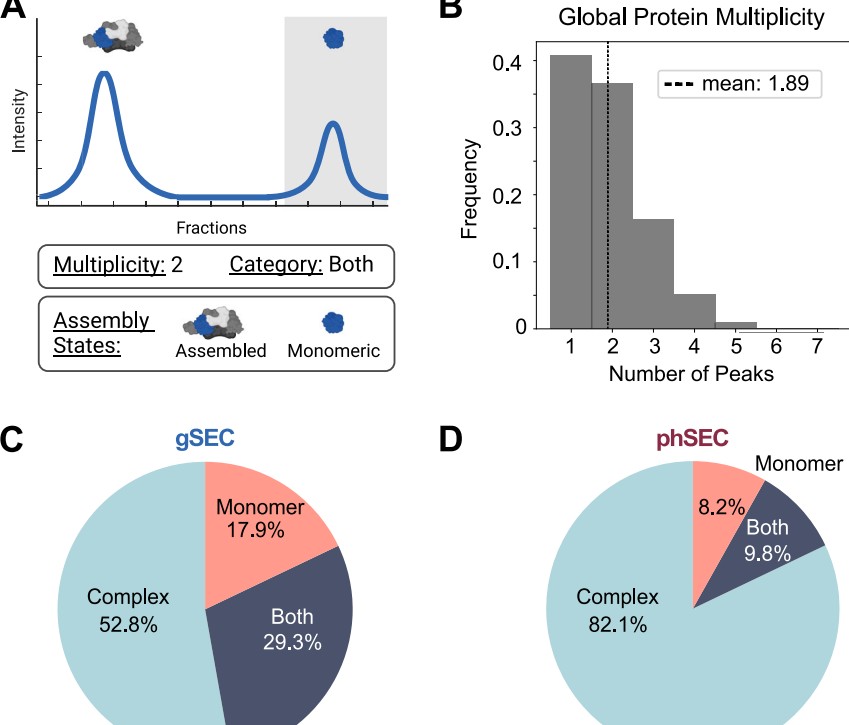

**Fig. 3 | A protein-centric analysis framework to study assembly states.**
**A** Hypothetical SEC-MX elution traces depicting a protein with multiple (2) assembly-states, each represented as a distinct peak. Peaks are categorized as monomeric or assembled based on molecular weight (MW) estimation (detailed in the Methods section). The MW range for monomer elution is depicted as a gray shaded area. Created in BioRender. Jovanovic, M. (2024). [https://BioRender.com/ s07v514][77]. **B** Frequency distribution of peak multiplicity per protein shows that >50% of proteins elute with >1 peak. **C, D** Pie charts showing the percentage of proteins eluting exclusively as monomers, exclusively as complexed, or both. Peaks were identified based on peptide data and counted per protein. gSEC in C and phSEC in D.

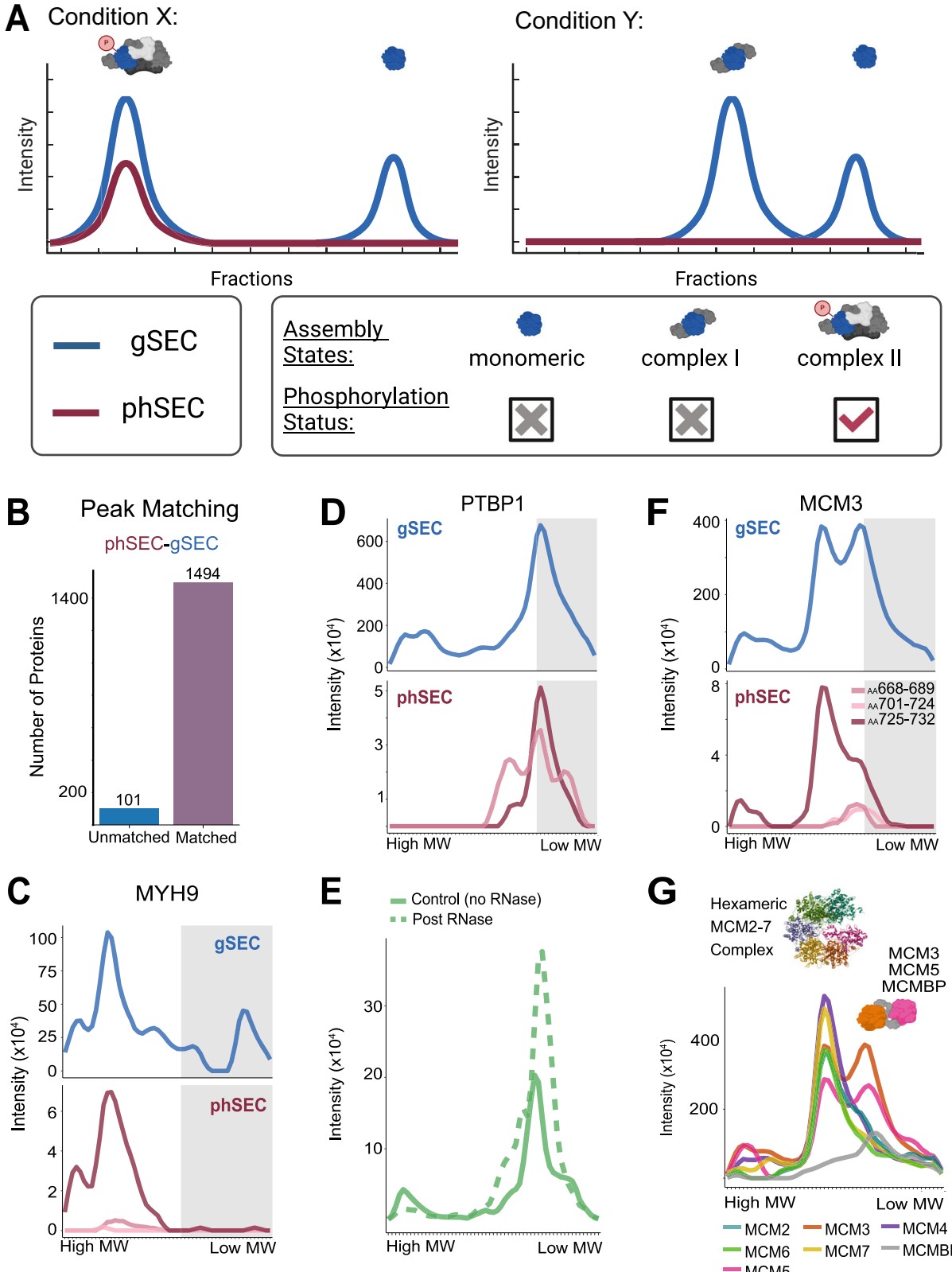

BioPlex, providing an additional comparison of the protein assembly states in HEK293 and HCT116 based on an orthogonal method.

To set up the differential comparison between HEK293 and HCT116, we multiplexed the same set of SEC fractions from each cell line in the same TMT mix (Fig. 5B, Supplementary Fig. 1B). This resulted in nearly 100% overlap in peptide coverage between the biological samples in both datasets (gSEC and phSEC), indicating that the

absence of a peak in one condition is unlikely due to coverage differences and can be more confidently interpreted as a difference in assembly state. Overall, the SEC profiles of HEK293 and HCT116 were highly correlated (Fig. 5C), identifying that 78% of gSEC assembly states and ~68% of phSEC assembly states were shared in both cell lines (Fig. 5D). Identifying the assembly states exclusive to each cell line's dataset opens many possibilities for exploring potential interaction-

**Fig. 4 | Mapping PTMs onto assembly-states. A** Hypothetical SEC-MX elution traces depicting how protein elution can differ between conditions. A single phosphorylated protein with multiple assembly-states is depicted. Under condition X, the gSEC traces suggest elution as a monomer (not phosphorylated) and as a large complex that co-elutes with a phosphopeptide in phSEC. Under condition Y, the monomer peak is still evident, while the different position of the larger sized elution peak suggests that the protein is taking part in a different complex, with no evidence for phosphorylation. Created in BioRender. Jovanovic, M. (2024). [https://BioRender.com/ s07v514][77]. **B** Number of proteins for which phSEC peaks were matched to the gSEC peaks, based on their apex position: 1494 out of the 1595 (94%) proteins measured in both gSEC and phSEC had a least one phSEC peak that matched to a gSEC peak. Note: one protein was dropped from the analysis (as compared to the number reported in Fig. 2B), due to low signal/noise ratio. **C** SEC-MX elution traces in gSEC (top) and phSEC (bottom) for MYH9 in HEK293 cells (averaged across replicates). Gray boxes indicate the range of fractions covering

the monomeric form of the protein. The three pink shaded traces in phSEC represent 3 different phosphopeptides all spanning the phosphorylation site on Serine-1943. **D** Same as C, for PTBP1. The two pink shaded traces in phSEC represent 2 different phosphopeptides spanning the phosphorylation sites on Serine-140 (localization probability = 0.5) or Serine-141 (localization probability = 0.5). **E** PTBP1 elution in SEC-DIA performed on HEK293 cell lysates before (solid line) or after (dashed line) RNAse treatment. **F** Same as C, for MCM3. The three pink shaded traces in phSEC represent 3 different phosphopeptides spanning different amino-acid (AA) sequences as indicated. **G** gSEC elution traces for all MCM2-7 complex subunits (HEK293, averaged across replicates) support the existence of two assembly-states: the full complex and an intermediate assembly-product comprising of MCM3-MCM5-MCMBP. Crystal structure of human single hexameric MCM2-7 complex is shown (PDB 7W68, deposited by Xu, N.N. et al., 2021-12-01[78]). The hypothetical schematic of the MCM3-MCM5-MCMBP complex was Created in BioRender. Jovanovic, M. (2024). [https://BioRender.com/ s07v514][77].

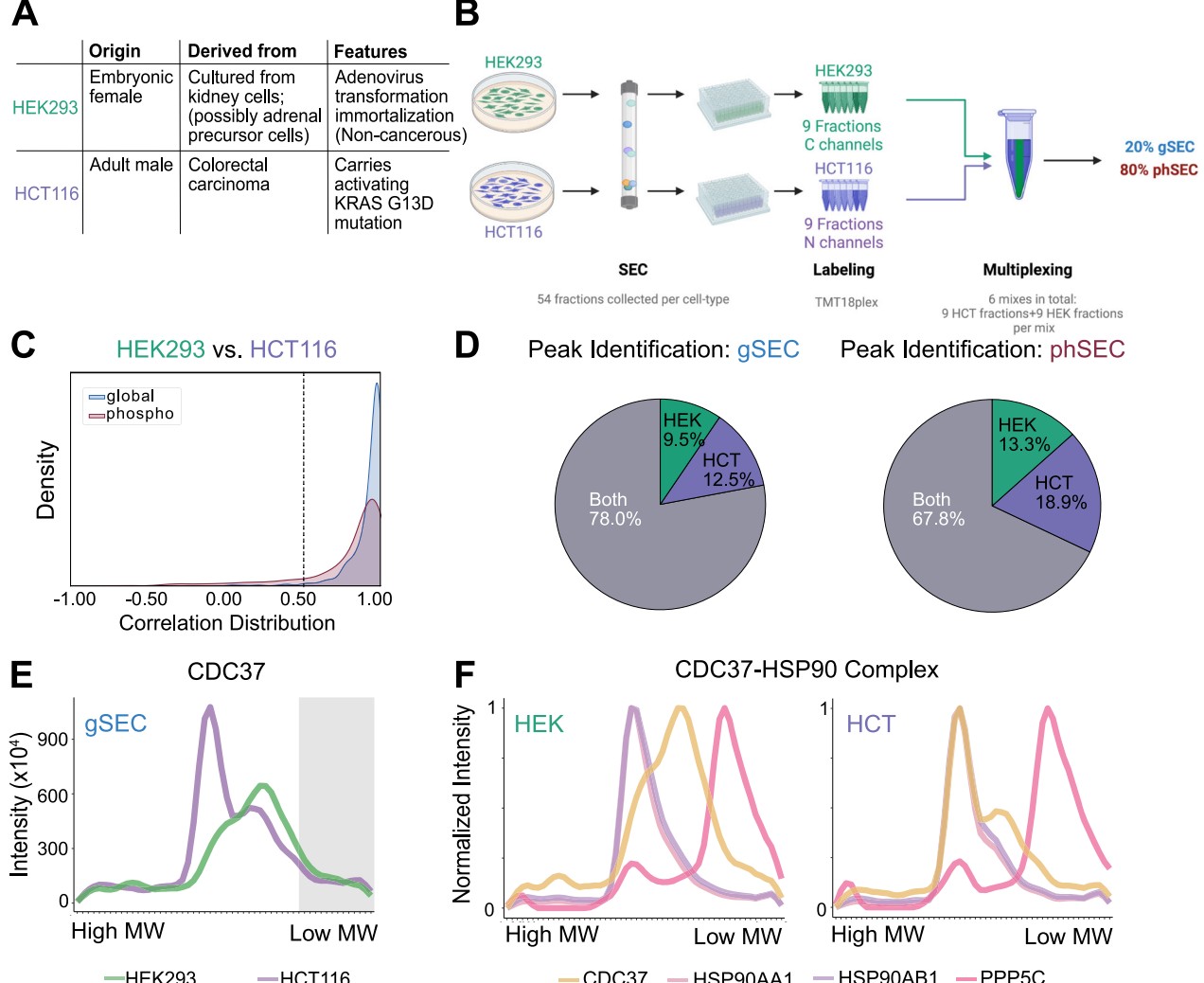

**Fig. 5 | SEC-MX enables differential comparison between biological samples. A** Summarized table of differences between HEK293 and HCT116 cell lines. **B** Overview of experimental multiplexing workflow. The same 9 fractions from HEK293 or HCT116 cells were multiplexed together, resulting in 6 combined mixes. After pooling, 80% of each mix were allocated for phosphopeptide enrichment and measurement of phSEC, while the remaining 20% were processed for gSEC. Created in BioRender. Jovanovic, M. (2024). [https://BioRender.com/ s07v514][77]. **C** Distribution of Pearson correlation coefficients calculated between HEK293 and HCT116 peptide elution profiles for the intersecting peptides in gSEC (blue) and

phSEC (maroon). **D** Pie charts showing the percentage of peaks identified exclusively in the HEK293 dataset, the HCT116 dataset, or mutual, in gSEC (left) and phSEC (right) based on the peptide-level data. **E** SEC-MX elution traces in gSEC for CDC37 (averaged across replicates). HEK293 (green) and HCT116 (purple). The gray shaded area indicates the range of fractions covering the monomeric form of the protein. Fold-change of peak heights (HCT116/HEK293) is 3.5 in fraction 24, and 0.8 in fraction 33. **F** Normalized SEC-MX elution profiles for CDC37-HSP90 complex subunits (averaged across replicates) in HEK293 (left), HCT116 (right). Traces are color-coded by subunit, as indicated under the plot.

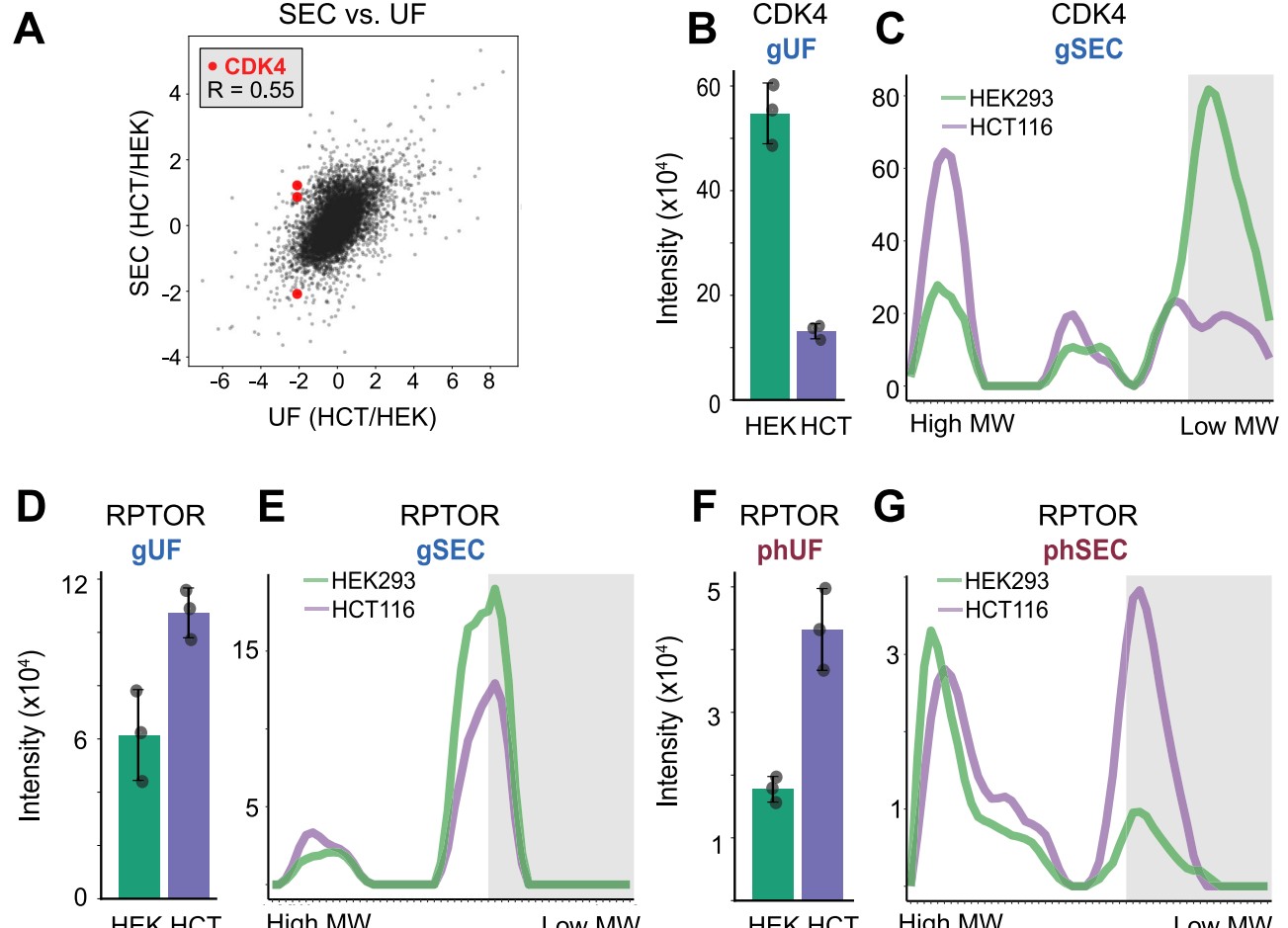

**Fig. 6 | Assembly state resolution unveils differential analysis beyond abundance measurements. A** Scatterplot comparing HCT116/HEK293 ratios in gSEC (peak heights) versus gUF (total intensity), R = 0.55 using a linear regression model. Each dot represents a protein assembly-state. CDK4 is highlighted in red. See Supplementary Fig. 6 for further comparisons between SEC and UF. **B** Barplot (mean +/- standard deviation across replicates) of CDK4 intensity in gUF ($p = 0.0003$, two-tailed unpaired $t$-test, overlaid data-points represent values measured in individual replicates). **C** gSEC elution traces for CDK4 (averaged across replicates). HEK293 (green) and HCT116 (purple). Gray boxes indicate the range of fractions covering the monomeric form of the protein. **D** Same as B, for RPTOR ($p = 0.015$, two-tailed unpaired $t$-test). **E** Same as C, for RPTOR. **F** Same as B, for RPTOR in phUF ($p = 0.0029$, two-tailed unpaired $t$-test). **G** Same as C, for RPTOR in phSEC.

rewiring events. For a full list of proteins identified with distinct assembly states in each dataset, please refer to Supplementary Data 3.

Moving forward, we compared our differential assembly state data to BioPlex. First, we took a closer look at BioPlex network complexes that showed a high degree of shared interactions between HEK293 and HCT116. In agreement with the BioPlex network, we found that the ubiquitously-expressed complexes Ribonuclease P and the COP9 Signalosome were intact in both cell lines (Supplementary Fig. 5A, B). In contrast, the interactions that connect GINS-CDC45 to MCM to form the eukaryotic replicative DNA helicase CDC45-MCM-GINS (CMG), were more readily purified from HEK293 cells as compared to HCT116 in BioPlex[44]. The authors speculated that this might be attributed to differential abundance of the complex subunits, rather than reflect rewiring of the complex. In SEC-MX, we indeed found that CMG subunits eluted together in both cell lines (Supplementary Fig. 5C), and found that the GINS subunits (GINS1-4) were more abundant in HEK293 over HCT116 (Supplementary Fig. 5D) – confirming the BioPlex authors' hypothesis.

Unlike IP-based methods such as BioPlex, SEC-MX allowed us to identify differences between assembly states. For example, SEC-MX detected a differential pattern in the association of the Hsp90 co-chaperone Cdc37 (CDC37) with the HSP90 chaperone, which BioPlex did not observe (Supplementary Fig. 5E). We found that CDC37 was

more highly expressed in HCT116 cells and that its distribution between two alternative assembly states differed between the cell lines (Fig. 5E), suggesting greater recruitment of CDC37 to the HSP90 chaperone in HCT116 compared to HEK293 cells (Fig. 5F). Notably, CDC37 overexpression is a hallmark of cancer, and its recruitment to the HSP90 complex promotes oncogenesis, making the disruption of HSP90–CDC37–kinase interactions a potential target for cancer therapies[46]. This differential CDC37 assembly state likely reflects the different origins of the cell lines, as HCT116 is derived from colorectal carcinoma, while HEK293 is not of cancer origin.

Together, these examples underscore the strength of SEC-MX in detecting differences between multiple cell lines – HEK293 and HCT116. Our method provided an orthogonal tool to BioPlex for examining protein complexes, while also adding assembly state resolution.

## Assembly state resolution unveils differential analysis beyond abundance measurements

Motivated to explore the quantitative power of SEC-MX, we compared assembly states aligned between both conditions by calculating the peak-ratios (HCT116/HEK293). To highlight the advantages of the assembly state resolution enabled by SEC-MX, we compared the assembly state ratios to the total abundance ratios between HCT116

and HEK293 cells in unfractionated samples (global – gUF; enriched phosphopeptides – phUF, provided in Supplementary Data 4). We observed a strong correlation between SEC-MX and unfractionated data in the HCT116/HEK293 ratios at both the global protein and phosphopeptide levels (Fig. 6A, Supplementary Fig. 6A–C). Notably, proteins that deviated from this overall correlation were often those with multiple assembly states, each exhibiting a distinct HCT116/HEK293 ratio. One example of this is the cell-cycle regulator Cyclin dependent kinase 4 (CDK4). CDK4 total abundance was shown to be upregulated in HEK293 compared to HCT116 cells in BioPlex[44] – a finding we recapitulated in gUF ($p = 0.0003$, two-tailed unpaired t-test, Fig. 6B). This is expected due to the upregulation of TP53 in HEK293[47] (shown in BioPlex and in our data, Supplementary Fig. 6D) – a result of the adenoviral transfection used to immortalize these cells – which in turn leads to the disassembly of the DREAM complex that inhibits CDK4 transcription[44]. However, a closer inspection of CDK4 gSEC traces shows that the assembly state that is upregulated in HEK293 cells is its monomeric form, which was shown to be inactive[48]. Meanwhile, we found two additional CDK4 assemblies that were downregulated in HEK293 compared to HCT116 (Fig. 6C), leading to the hypothesis that although the overall abundance of the protein is higher in HEK293, its activity might not necessarily be higher. These findings showcase how SEC-MX not only provides a nuanced view of protein assembly states but also reveals insights into functional regulation that may be overlooked by total abundance measurements, offering a more accurate reflection of protein activity between cell lines.

Altogether, SEC-MX not only provides assembly state resolution but also allows us to annotate each assembly state with phosphorylation data. While fold changes at the gSEC level address differences in assembly state abundance, phSEC fold changes offer a measure of the increased or decreased phosphorylation of individual assembly states. A relevant example in HEK293 and HCT116 cells is the regulatory-associated protein of mTOR (RPTOR). A phosphopeptide spanning RPTOR Serin-863 was identified in both the phUF and phSEC datasets. It was enriched in HCT116 in the phUF dataset ($p = 0.0029$, two-tailed unpaired t-test, Fig. 6D–G). However, the SEC-MX dataset revealed that only the monomeric form of RPTOR was more highly phosphorylated in HCT116 (Fig. 6G), whereas the assembled state (co-eluting with other mTORC1 subunits, Supplementary Fig. 6E) showed no differential phosphorylation. Previous observations in the literature showed that phosphorylation of RPTOR Serine-863 enhances mTORC1 kinase activity towards EIF4EBP[49]. However, it was assumed that this effect is observed in the context of mTORC1. These findings highlight the need for follow-up experiments to elucidate the role of differential phosphorylation of RPTOR Serine-863 in the monomeric form. Together, these findings illustrate how SEC-MX enhances our understanding of the interplay between protein assembly states and phosphorylation, highlighting its potential to uncover hypotheses of functional regulation that may be obscured by traditional abundance measurements.

## Using SEC-MX to explore functional pathways and provide molecular insights

As shown above, SEC-MX enables the enrichment of phosphorylated assembly states – providing quantitative information on the differential expression of protein assembly states (gSEC) and their associated phosphorylation (phSEC) at assembly state resolution. Moving forward, we aimed to explore how the multiple dimensions of SEC-MX data can be used to systematically investigate functional pathways on a larger scale. One aspect of this investigation was to distinguish assembly state regulation at the abundance level from regulation at the phosphorylation level. For example, an observed increase in the phosphorylation of a protein can imply different modes of regulation whether or not a corresponding increase was observed in the protein abundance. Therefore, we compared the HCT116/HEK293 peak ratios

between phSEC and gSEC, as presented in the scatterplot in Fig. 7A. We then classified differential assembly states in the gSEC and/or phSEC datasets using a cutoff of a 2-fold difference in peak ratio. These cutoffs divided our data to 9 response groups based on whether the assembly states differed between HEK293 and HCT116 at the global level, phosphorylation level, or neither (Fig. 7A). In-line with our previous analyses, we observed that most proteins (781) had assembly states which were not significantly changed at either level (group E: gNS/phNS; NS stands for "Not Significant"). Interestingly, the next largest group was of proteins with assembly states upregulated at the phosphorylation level, without a significant change in assembly state abundance (groups B: gNS/phHCT(up) with 305 proteins, and H: gNS/phHEK(up) with 224 proteins, Fig. 7B). Many fewer proteins had phosphorylated assembly states that passed the cutoff in gSEC but not in phSEC (38 in HEK293 and 31 in HCT116), and a total of 4 proteins had gSEC and phSEC ratios in opposite directions. Of note, proteins with multiple assembly states can be represented in multiple distinct groups (full lists of proteins represented in each group are provided as Supplementary Data 5).

To decipher whether the distinct response groups we identified by SEC correspond to specific functional pathways, we conducted pathway over-representation analyses against the Reactome database[50,51] (Supplementary Fig. 7A). Additionally, given we identified distinct groups enriched in phosphorylation but not necessarily total protein changes – we hypothesized that each group may be regulated by different kinases and conducted kinase-target enrichment analysis[52] (Supplementary Fig. 7B). These analyses, as expected, showed an enrichment of p53-related pathways from proteins with increased phosphorylation in HEK293 cells (Fig. 7A, highlighted in pink). Similarly, HEK293 phosphorylation was also enriched with targets of the kinases ATR and ATM – activators of DNA damage signaling[53] (Supplementary Fig. 7B). In contrast, phosphorylated assembly states in HCT116 were enriched with components of the mTOR complex 1 (mTORC1) mediated signaling (Fig. 7A, highlighted in yellow), including kinase targets of the ribosomal protein S6 kinase beta-1 (RPS6KB1) and mTOR itself (Supplementary Fig. 7A, B). The differential mTOR signaling between HEK293 and HCT116 cells may be due to the activating G13D mutation in the GTPase KRas (KRAS) in HCT116[45], which leads to its accumulation in an active state, causing persistent activation of both the RAF/RAS/MAPK and PI3K/AKT/mTOR pathways[45,54–57]. In summary, our findings demonstrate how the distinct response groups identified by SEC-MX enable the detection of alternative pathway regulation in each cell line.

Aside from uncovering differential pathways through term-enrichment, as shown above, SEC-MX can annotate widespread regulatory pathways with expression and phosphorylation patterns. To exemplify this, we annotated the signaling networks around two pathways found in the prior scatterplot: mTORC1[51,58,59] and TP53[51,60,61] (Fig. 7A). Both pathways are central hubs in the cell signaling network, and are regulated in response to various cellular conditions. TP53 responds primarily to DNA damage, and cell-cycle state. MTOR integrates multiple signaling cascades that sense cell homeostasis including the availability of essential amino acids, growth factor stimulation, and more. Although TP53 and MTOR are responding to distinct cellular conditions, there is also crosstalk between these pathways[62]. Both involve multiple upstream effectors – many of which affect phosphorylation of pathway members and/or their physical interactions. Since SEC-MX detected many effectors in these pathways, we show the results obtained for some of the more prominent proteins (Fig. 8). This analysis reveals the detail of differential activity provided to each protein; some with multiple phosphorylation sites and others with multiple assembly states that are distinctly regulated. This analysis highlights the depth of coverage and the detailed information contents in SEC-MX, delineating it as a valuable tool to study the molecular details of differential pathway regulation.

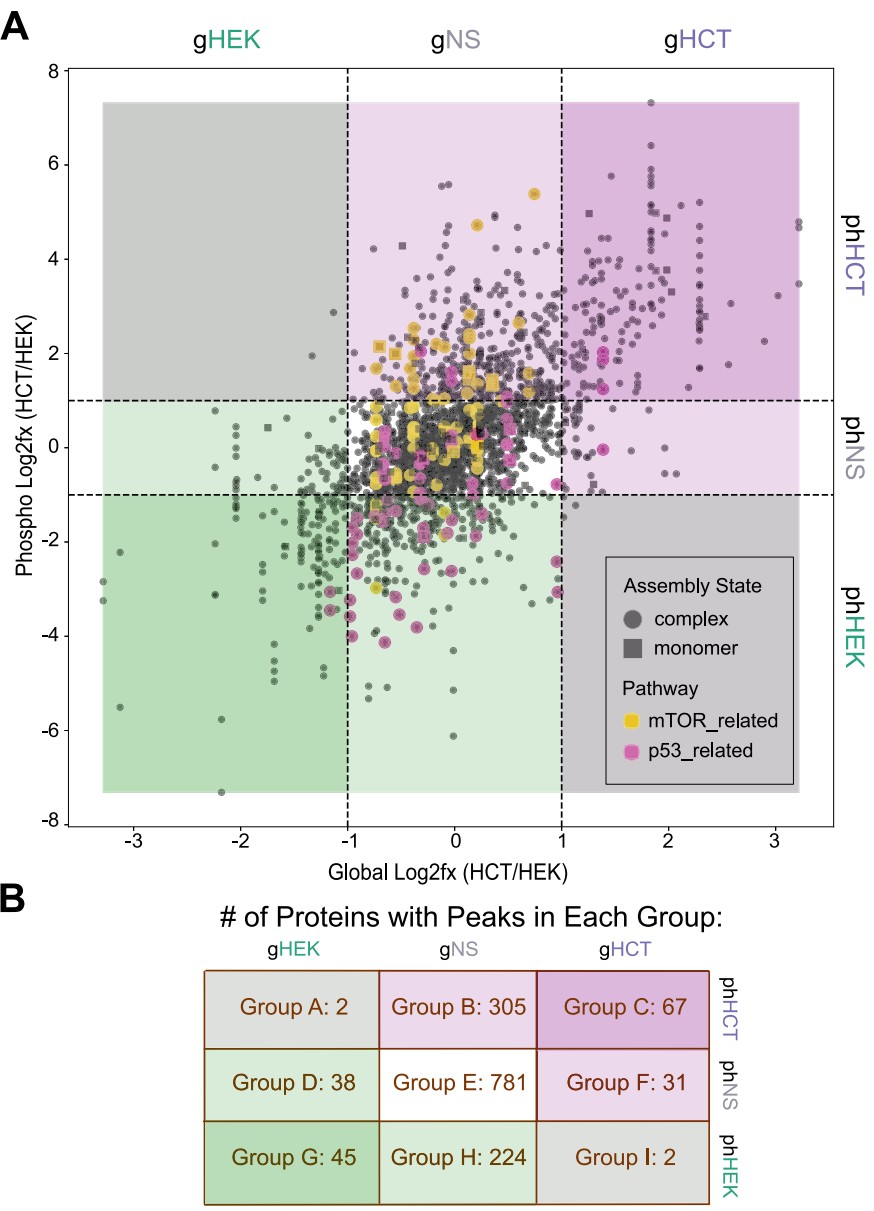

**Fig. 7 | SEC-MX enables exploration of functional pathways. A** Scatter plot comparing the log2 ratio of peak heights (HCT116/HEK293) in phSEC (y-axis) and gSEC (x-axis), based on the subset of peaks aligned in both cell lines. Dashed lines indicate the +/- $\log_2 (2)$ cutoff for each dimension, dividing the plot into 9 groups based on assembly states differences at the abundance and/or phosphorylation levels, as indicated on the top and right sides of the scatter. Proteins involved in the "mTORC1 mediated signaling" pathway (Reactome R-HSA-166208), and MTOR and RPS6KB1 kinase targets are colored in yellow. "Transcriptional Regulation by TP53" pathway (Reactome R-HSA-3700989) and CHEK2 and ATR kinase targets are colored in magenta. **B** Table indicating the number of proteins with peaks in each of the nine groups described in 7A.

In conclusion, we used our peak-focused approach to analyze differences in assembly states between biological conditions, both at the protein (gSEC) and at the phosphopeptide level (phSEC). Although most assembly states are consistent between the two cell lines, our results revealed proteins with assembly states exclusive to each cell line's dataset, as well as proteins with differential assembly state enrichment at both the global and phosphorylation levels. Furthermore, we identified cases of differential phosphorylation between distinct assembly states of the same protein, highlighting the need to define the specific assembly states of a phosphorylated protein to begin exploring the potential functional outcomes of these modifications. We showed how such a detailed analysis can generate testable hypotheses on the regulatory connections between phosphorylation and assembly states. Lastly, we also exemplified how SEC-MX provides this assembly information at great depth, covering a high percentage

of all proteins in pathways of interest (as exemplified by the mTOR pathway; Fig. 8). Together, our results support the utility of SEC-MX and the peak-focused analysis approach in providing valuable insight on the dynamics of assembly state regulation (abundance and phosphorylation) between conditions for thousands of proteins, with molecular details beyond what is detected by analysis of expression levels alone.

## Discussion

In this study, we developed SEC-MX, a multiplexed SEC-MS method that allows for the mapping of PTMs to assembly states and enables differential quantification between conditions. We showed that SEC-MX performs as well as state-of-the-art label free SEC-MS methods in terms of coverage and PPI identification, and in a fraction of the needed LC-MS/MS runs (Fig. 1). Furthermore, we showed the key value of

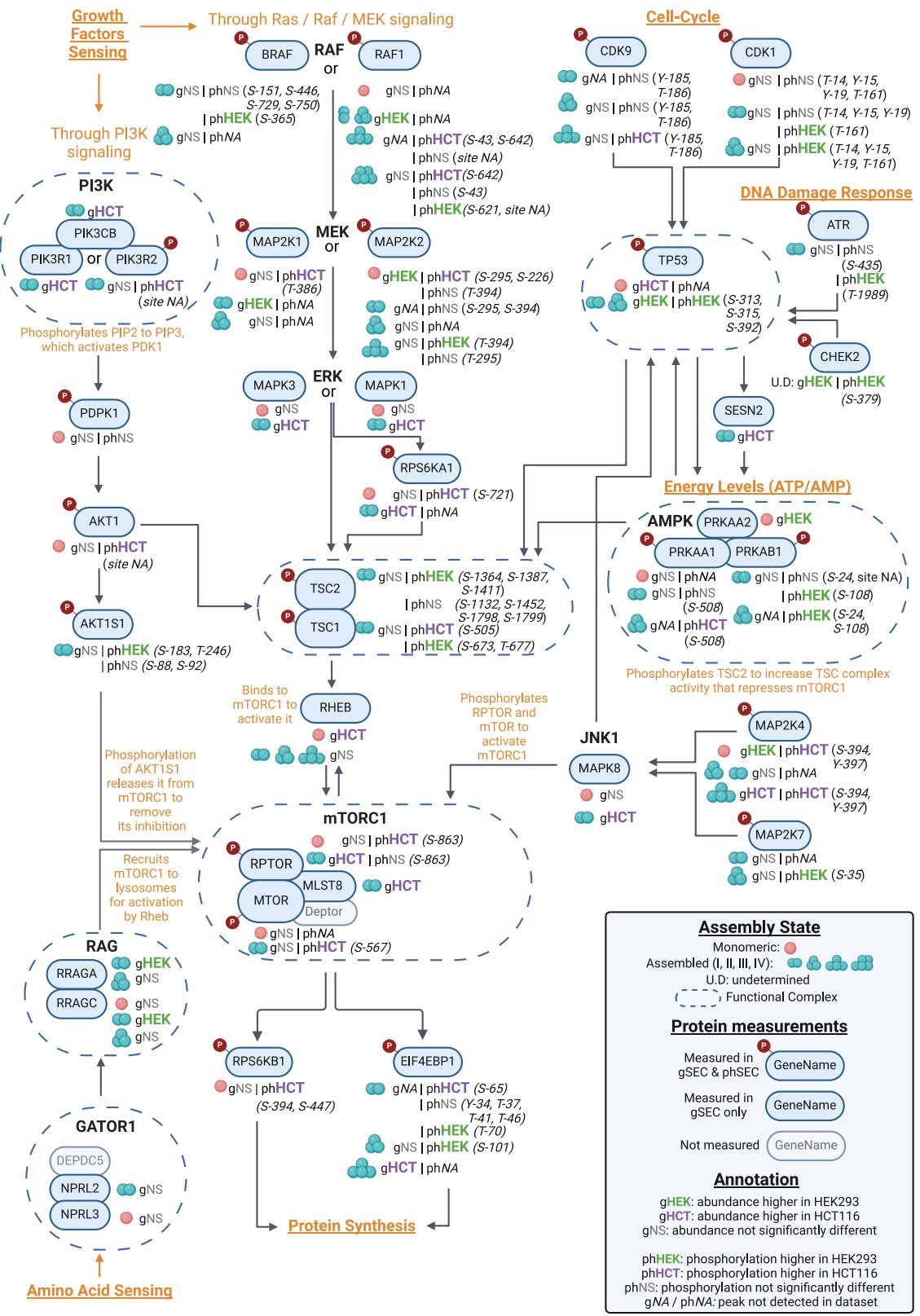

**Fig. 8 | Using SEC-MX to provide molecular insights.** Representation of selected proteins from the pathways mentioned in Fig. 7, based on published literature[50,57–61]. SEC-MX patterns are summarized on each protein: gHEK, gHCT, and gNS labels indicate assembly state abundance is higher in HEK293, HCT116, or not significantly changed, respectively. phHEK, phHCT, and phNS labels similarly indicate assembly state phosphorylation. gNA / phNA indicates a protein / phosphopeptide was measured but no peak was detected for the specific assembly state. Numbers in brackets indicate the phosphorylation site identified (S – Serine, T- Threonine, Y – Tyrosine), site NA indicates a phosphopeptide was measured but no site was identified with localization probability > 0.9 and therefore the specific site is not reported. Created in BioRender. Jovanovic, M. (2024). [https://BioRender.com/s07v514][77].

SEC-MX by enriching for phosphopeptides in the fractionated SEC samples. Using SEC-MX, we generated a dataset comprising the SEC elution profiles of thousands of phosphopeptides and their corresponding parent-proteins (Fig. 2). We displayed how a careful multiplexing design can position SEC-MX as especially well-suited for quantifying differences between samples. Additionally, we analyzed these datasets with a focus on peak-level comparisons to overlay PTM information on assembly states and/or compare them between biological conditions.

Our dataset provides a systematic view of the interplay between phosphorylation events and assembly states, highlighting proteome dynamics beyond abundance changes. In-line with previous reports[1,2,15], we confirmed that the large majority of the proteins in the cell are assembled with other molecules. Building on this concept, our analysis reveals that over half of the measured proteins eluted in multiple assembly states, with PTMs often associated with only a subset of these states, and potentially changing between conditions (Fig. 3). We further exemplified how our approach can generate testable hypotheses regarding the relationships between phosphorylation events and protein complex assembly (Fig. 4), and how they may differ between conditions (Figs. 5–8). For example, we showed the importance of considering the assembly state in which a protein is differentially regulated in order to investigate its downstream effect (CDK4, RPTOR).

In developing our protein-centric analysis approach, we decided to deviate from state-of-the-art protein correlation profiling (PCP) that aims to uncover pair-wise PPIs (Prince[63], EPIC[24], SECAT[22,64]). Inspired by complex-centric programs such as CCprofiler[15] and PCprophet[33], we defined distinct assembly states based on the elution fractions of SEC peaks. We then focused on how individual protein assembly states change between samples, similar to other protein-centric approaches[2]. Our analysis framework proved to be a simple and comprehensive tool to compare assembly states between conditions while overlaying PTMs, which we provide as an open resource (see Code Availability). In practice, we identified assembly states for nearly every analyzed protein/peptide, as opposed to methods focused solely on the complex/PPI level, which often result in high-confidence networks that cover only a fraction of all measured proteins. However, future applications of SEC-MX analyses should attempt to tie-in PPI networks to provide statistical evidence to the composition of different assembly states.

Similar to SEC-MX, other techniques apply a protein-centric approach to detect dynamic changes in the functional proteome, such as thermal proteome profiling (TPP[65,66]), covalent protein-painting (CPP[67]) and limited-proteolysis mass spectrometry (LiP-MS[3]). These methods are sensitive in detecting protein functional changes that can result from a wide array of mechanisms, including and not limited to, molecular interactions and PTMs. However, these methods do not provide individual assembly state resolution – limiting their ability to identify sub-complex mechanisms without detailed biochemical follow-ups.

Moreover, co-fractionation analysis following treatment with non-specific phosphatases (phospho-DIFFRAC[28]) also took a protein-centric approach to identify proteins with altered SEC elution profiles pre/post-treatment. Phospho-DIFFRAC provides general causality on the effects of phosphorylation on assembly states – measuring the effects of phosphatases. However, it lacks direct measurements of phosphopeptides and cannot shed light on the specific modified sites. For example, both our study and phospho-DIFFRAC suggested that the assembly state of MYH9 is regulated by phosphorylation. Yet, SEC-MX was able to pinpoint the specific phosphorylated peptide, which contained a modification site known to affect interactions. While SEC-MX cannot determine causality in the relationships between PTMs and assembly states, we showed how it can serve as a valuable hypothesis-generating tool, guiding the design of follow-up studies into the causal effects of specific PTM sites. Furthermore, by developing the

multiplexing schemes, SEC-MX could facilitate the collection of detailed time-course measurements to validate these hypotheses.

In its current form, SEC-MX depends on the use of TMT isobaric tags, which result in compression of relative quantifications[68,69]. However, our proposed protein-centric peak analysis partially mitigates this issue by applying within-protein and within-mix comparisons. In addition, since TMT requires data-dependent-acquisition (DDA), which potentially reduces the depth of coverage, it can result in missing data points. Therefore, before claiming that a peptide does not have a peak in a certain region, special attention should be made to verify that it was measured in the given TMT-mix. To overcome these issues, future development of SEC-MX should test the use of isotopic tags compatible with DIA methods, such as mTRAQ[70], as they are expected to become available in >3 channels. In addition, labeling reagents such as TMT are often costly and require additional experimental steps (i.e., more time and expertise) compared to label-free analyses, which should be taken under consideration in experimental design and planning.

Lastly, it should be noted that sample preparation and fractionation by SEC can take as much as 2-3 h for a given sample. Since weaker or more transient interactions are prone to dissociate during the process, SEC-MS is often limited to the study of stable complexes. To overcome this, previous SEC works have attempted to stabilize complexes by crosslinking[71,72]. This direction should be explored for compatibility with SEC-MX in future iterations of the method.

In conclusion, SEC-MX represents a significant advancement in the study of protein regulation and cellular functions by providing a comprehensive platform for the simultaneous characterization of post-translational modifications and assembly states. Our method not only streamlines measurement processes and enables PTM enrichment but also offers an approach to explore the intricate interplay between these molecular events. We provide a granular perspective on protein regulation by generating a proof-of-concept dataset encompassing the identification and quantification of assembly states and PTMs across two distinct biological samples. Moreover, our focus on distinct assembly states highlights the diverse regulatory processes of proteins, underscoring the necessity of analytical tools such as SEC-MX. While our study primarily examined phosphorylation for proof-of-concept, the versatility of SEC-MX allows for future exploration of additional modifications (e.g., acetylation), thereby broadening its applicability across various biological contexts. Ultimately, SEC-MX enables researchers to explore dynamic cellular processes and sheds light on the intricate mechanisms driving protein complexity and regulation.

## Methods

### Cell culture

HEK293XT (HEK293) cells (Takara Bio Lenti-X 293 T, #632180) were provided by the Yeo lab at UC San Diego (SEC-DIA versus SEC-MX experiments), or purchased from ATCC (ATCC, CRL-3216, in HEK293 versus HCT116 experiments), HCT116 cells were provided by the Prives lab at Columbia University (originally contributed by Bert Vogelstein laboratory at Johns Hopkins University). HEK293 cells were cultured in DMEM (containing L-glutamine and Sodium pyruvate) and HCT116 in McCoy's 5 A media. Both media were supplemented with 10% Fetal Bovine Serum and Penicillin (100 U/mL) Streptomycin (100 µg/mL). Cells were grown to 80–90% confluency and were harvested at passages 6–20.

### Sample preparation for SEC

SEC sample preparation was as previously described in Bludau et al. 2020[16]. Cells (25-40 million per sample) were harvested by scraping in ice cold PBS, washed and pelleted. Pellets were flash frozen in liquid nitrogen and stored in -80 °C. Upon thawing, cell pellets were lysed in cold lysis buffer (for TMT versus DIA comparisons: 150 mM NaCl,

50 mM Tris pH 7.5, 1% IGPAL-CA-630, 5% Glycerol; for HEK-HCT experiments: 50 mM HEPES pH 7.5, 150 mM NaCl, 0.5% NP40) supplemented with 50 mM NaF, 2 mM $Na_3VO_4$, 1 mM PMSF, and 1X protease inhibitor cocktail (Sigma), followed by 10-30 min incubation on ice with intermittent vortexing. Cell lysates was then pre-cleared by 10 min centrifugation at 10,000 $g$ (4 °C) followed by 20 min of ultracentrifugation at 100,000 $g$ (4 °C). To dilute detergents in the buffer, samples underwent buffer exchange on Amicon® ultra-0.5 centrifugal filter with 30 kDa molecular weight cutoff (Sigma) into 50 mM HEPES pH 7.5, 150 mM NaCl and 50 mM NaF in iterative steps of no larger than 1:3 dilutions. The final dilution ratio of the original lysis buffer to detergent free buffer was 1:50. The cell lysate was further cleared by 5 min of centrifugation at 17,000 $g$ (4 °C). The concentration of the supernatant was measured by Nanodrop spectrophotometer (Thermo Scientific) and adjusted to 20–50 mg/ml. 2 mg of lysate were loaded on the SEC column per run.

## Size exclusion chromatography
Size exclusion was conducted on an Agilent 1260 Infinity II system operated with Agilent OpenLAB ChemStation software (version C.01.09). 2 mg of cell lysate at 20–50 mg/ml were loaded onto a Yarra SEC-4000 column (Phenomenex 00H-4514-K0, 3 μm silica particles, 500 A pores, column dimensions: 300 × 7.8 mm) and fractionated in SEC running buffer (50 mM HEPES pH 7.5, 150 mM NaCl) at a flow rate of 1 ml/min (first TMT experiment) or 0.5 ml/min (all other experiments) and 100 μL fractions were collected between minutes 6.5 to 16 or 11 to 30, respectively into 96 Well DeepWell Polypropylene Microplates (Thermo Scientific).

## Protein digestion and desalting
Following SEC fractionation protein concentration was measured using the Pierce™ BCA protein assay kit (Thermo Scientific) based on the manufacturer's instructions. Equal volumes ( ~ 80 μL) from each of the fractions containing proteins (54–72) were subsequently processed. Proteins were denatured by incubation with an equal volume of urea buffer containing 8 M urea, 75 mM NaCl, 50 mM HEPES (pH 8.5) and 1 mM EDTA at 25 °C, 600 rpm for 20 min in 96 Well DeepWell Polypropylene Microplates (Thermo Scientific). Proteins were then reduced with 5 mM DTT at 25 °C, 600 rpm for 45 min and then alkylated with 10 mM iodoacetamide (IAA) at 25 °C, 600 rpm for 45 min in the dark. Proteins were then diluted in a ratio of 1:3 with 50 mM HEPES (pH 8.5) to lower the urea concentration less than 2 M, and digested with trypsin enzyme (Promega) at 25 °C, 600 rpm overnight using 1:50 (enzyme to protein) ratio. Digested peptides were acidified using formic acid and desalted on in-house packed C18 StageTips (two plugs) on top of 96 Well DeepWell Polypropylene Microplates as elaborated in Rappsilber et al., 2007[73]. For DIA measurements, dried peptides were reconstituted to a final concentration of 0.5 μg/μL with 3% acetonitrile/ 0.2% formic acid. For TMT labeling purposes dried peptides were reconstituted in 50 mM HEPES (pH 8.5). An aliquot of 0.2 mg of each non-fractionated sample (for HEK and HCT global protein expression analysis using DIA) was processed in a similar manner.

For the HEK-HCT dataset we used a direct labeling method. Following fraction selection based on BCA measurements, proteins were denatured by incubation at 95 °C, 600 rpm for 10 mins, followed by two cycles of 1 min bath sonication. After samples cooled down to room temperature, proteins were reduced with 5 mM DTT at 25 °C, 600 rpm for 45 min and then alkylated with 10 mM iodoacetamide (IAA) at 25 °C, 600 rpm for 45 min in the dark. Proteins were then diluted in a ratio of 1:3 with 50 mM 4-(2-hydroxyethyl)-1-piperazinepropanesulfonic acid (EPPS), pH 9.0. pH was adjusted to ~ 8.2, and samples were subsequently digested with trypsin enzyme (Promega) at 25 °C, rpm 600 overnight using 1:50 (enzyme to protein) ratio, and TMT labeled following digestion.

## TMT labeling
For SEC-MX experiments used to compare to SEC-DIA samples were digested and desalted as elaborated above. The resulting peptides were reconstituted with 50 mM HEPES (pH 8.5). For direct labeling peptides were labeled in the adjusted digestion buffer (50 mM EPPS, pH adjusted to ~8.2). This protocol resulted in a mean labeling efficiency of 98.3%, (comparable to the classic protocol, mean efficiency of 98.7%), while minimizing loss of material due to multiple desalting steps.

For all samples, peptides were labeled by addition of TMTpro™ 18 plex reagents (Thermo Scientific) into the sample at a ratio of 1:3 (peptide to TMT) by mass in a final volume of 29% acetonitrile. The labeling reaction was incubated at 25 °C, 600 rpm for 1 hour before being quenched with a final concentration of 0.3% hydroxylamine. Samples were then pooled as described in the pooling scheme and dried at least half of the volume to lower the acetonitrile concentration to less than 5%. The labeled peptides were then acidified using formic acid (pH <3) and desalted on C18 StageTips (two plugs)[73]. The desalted peptides were dried and resuspended in 3% acetonitrile/ 0.2% formic acid for subsequent liquid chromatography-tandem mass spectrometry (LC-MS/MS) processing.

## Evaluating data compression in TMT experiments
Data compression was evaluated by comparing the HCT116/HEK293 ratios per protein between the TMT and DIA measurements. Since the TMT measurements consisted of SEC fractionated samples, signal intensity was first summed across all measured fractions to a single value per protein in each cell line/replicate. Replicates were averaged ($n = 2$ for gSEC-MX and $n = 3$ for gUF) and a single ratio (HCT116/ HEK293) was calculated per protein in the overlapping identifications between the datasets. Ratios were compared, resulting in TMT data compression of 69% (Supplementary Fig. 6B).

## Considerations for the choice of TMT multiplexing schemes
We initially attempted to perform IMAC enrichment on SEC fractions multiplexed using the "full overlap" scheme, resulting in a low number of recovered phosphopeptides ( ~ 1000 across all mixes). However, IMAC enrichment from SEC-MX mixes that were multiplexed with the "no overlap" scheme resulted in ~5 times more phosphopeptide identifications. Interestingly, comparing protein coverage in global datasets mixed with the full-overlap versus no-overlap schemes showed that protein coverage with the no-overlap scheme was only 10% lower, despite the 2-fold difference in measurements (4 versus 8 runs, for no-overlap versus full-overlap, respectively, Supplementary Fig. 1D, E). However, the no-overlap multiplexing scheme yielded only a third of the PPI identifications using SECAT (Supplementary Fig. 1F). Therefore, we concluded that while the "full-overlap" scheme is more suited for interaction analyses, the "no-overlap" scheme was preferable for subsequent enrichment of phosphorylated-peptides.

## Phosphopeptide enrichment using immobilized metal affinity chromatography (IMAC)
IMAC enrichment from pooled TMT labeled peptides was conducted following the protocol by Mertins and colleagues[32]. The mixes were as follows: 6 mixes per biological replicate containing 18 SEC fractions each; 9 from HEK293 cells and 9 from HCT116 cells, for a total of 54 SEC fractions from each cell line per replicate (see Supplementary Fig. 1B for the full labeling scheme). Ni-NTA Superflow Agarose beads (Qiagen, cat. no. 30410, 20 μl beads / 40 μl slurry per sample) were washed in water 3 times, then stripped from nickel by 30 min incubation with 100 mM EDTA, washed 3 times in water, and incubated with 10 mM iron (III) chloride in water for 30 min in room temperature. After 3 additional washes in water, iron coupled beads were resuspended in binding buffer (1:1:1 (vol/vol/vol) ratio of acetonitrile/methanol/0.01%

(vol/vol) acetic acid) in a final ratio of 1:1:1:1 beads/acetonitrile/methanol/0.01% (vol/vol) acetic acid. In parallel, TMT labeled peptides (post pooling) were resuspended to 0.5 μg/μl (~100 μg per sample, estimated based on BCA assay of the SEC fractions) in 80% (vol/vol) MeCN/0.1% (vol/vol) Trifluoroacetic acid (TFA). Eighty μl of bead slurry were added to the peptides solution and incubated for 30 min at room temperature. Following this binding step, supernatants were aspirated and the coupled beads were resuspended in 200 μl of 80% (vol/vol) acetonitrile /0.1% (vol/vol) TFA in order to be loaded on C18 stage tips for desalting. Two-plug C-18 stage-tips were conditioned twice with 100% Methanol, washed in 50% (vol/vol) acetonitrile in 0.1% (vol/vol) formic acid (FA), and equilibrated twice with 1% (vol/vol) FA. Then, the enriched beads were loaded onto the stage tips. Loaded beads were washed twice with 50 μl of 80% (vol/vol) acetonitrile/0.1% (vol/vol) TFA, then twice with 50 μl of 1% (vol/vol) FA. Phosphopeptides were trans-eluted from the beads to the C18 material by three iterations of 70 μl of agarose-bead elution buffer (192.5 mM monobasic potassium phosphate / 307.5 mM dibasic potassium phosphate). Stage tips were washed twice in 1% (vol/vol) FA, and peptides eluted using 60 μl of 60% (vol/vol) acetonitrile / 0.2% (vol/vol) FA. Eluted peptides were dried using a savant speedvac and reconstituted in 15 μl of 3% (vol/vol) acetonitrile / 0.2% (vol/vol) FA. Enrichment yield was estimated by calculating the number of phosphorylated peptides over the total peptides identified per mix and is 96.6 +/- 2.4% (mean +/- stdev), see Supplementary Fig. 3A.

## LC-MS/MS

LC-MS/MS analysis was performed on a Q-Exactive HF. 5 μL of total peptides (at estimated 0.5 μg/μL) were analyzed on a Waters M-Class UPLC using a C18 Thermo EASY-Spray column (2um, 100 A, 75um x 25 cm, or 15 cm) or IonOpticks Aurora ultimate column (1.7 um, 75 um x 25 cm) coupled to a benchtop ThermoFisher Scientific Orbitrap Q Exactive HF mass spectrometer. Peptides were separated at a flow rate of 400 nL/min using solvents A (0.1% formic acid in water) and B (0.1% formic acid in acetonitrile) with the following gradients of solvent B: for SEC-DIA a linear 20 min gradient from 2% to 14%, followed by a linear 15 min gradient from 14 to 22%, followed by a linear 5 min gradient from 22% to 30%, 2 min gradient from 30 to 60% solvent B, for DIA runs for unfractionated samples and SEC-MX runs a linear 4 min gradient from 5% to 8%, followed by a linear 85 min gradient from 8 to 22%, followed by a linear 20 min gradient from 22% to 30%, 14 min gradient from 30 to 60%. Each sample was run for either 160 min (DIA runs for unfractionated samples), 150 min (SEC-MX), or 70 min (SEC-DIA), including sample loading, column washing and equilibration times. For DIA runs MS1 Spectra were measured with a resolution of 120,000, an AGC target of $5 \times 10^6$ and a mass range from 350 to 1650 m/z. 63 isolation windows of 20 m/z were measured at a resolution of 30,000, an AGC target of $3 \times 10^6$, normalized collision energies of 22.5, 25, 27.5, and a fixed first mass of 200 m/z. For DDA runs MS1 Spectra were measured with a resolution of 120,000, an AGC target of $3 \times 10^6$ and a mass range from 300 to 1800 m/z. Top12 MS2 spectra were acquired at a resolution of 60,000, an AGC target of $1 \times 10^5$, an isolation window of 0.8 m/z, normalized collision energies of 27, and a fixed first mass of 110 m/z. Carryover of sample between consecutive LC-MS/MS runs was measured and found negligible.

The following samples were measured and analyzed: (1) SEC-MX (HEK293) - overall 8 TMT mixes in 2 biological replicates (16 LC-MS/MS runs); (2) SEC-DIA - overall 57 fractions per dataset, in 2 conditions (plus/minus RNase) and 2 biological replicates (228 LC-MS/MS runs); (3) SEC-MX for HEK293 and HCT116 - 12 mixes for gSEC with full overlap (one replicate), 6 mixes for gSEC with no overlap (one replicate), 6 mixes for phSEC in 2 biological replicates (30 LC-MS/MS runs); (4) unfractionated dataset - HEK293 and HCT116 gUF in 3 replicates each, HEK293 and HCT116 phUF in 3 replicates each (12 LC-MS/MS runs).

## Data analysis

**Searches.** Proteomics raw data were analyzed using the directDIA method on SpectroNaut v16.0 for DIA runs or SpectroMine (3.2.220222.52329) for DDA runs (Biognosys). Reference proteome used was human UniProt database (Homo sapiens, UP000005640). For SEC-MX, search parameters were set to BGS factory settings for TMTpro 18 channels, modified without automatic cross-run normalization or imputation. These parameters were as follows: digestion enzyme – Trypsin (cleavage sites after Arginine or Lysine), maximum missed cleavages 2, carbamidomethyl as fixed modification across all searches, TMTpro 16 as fixed modification (lysine and protein N-term) when applicable, acetyl (protein N-term) and Oxidation (Methionine) as variable modifications across all runs, phospho(STY) as variable modificaiton when applicable, mass tolerance for precursor and fragment ions was set as "dynamic" (see Biognosys documentation), peptide maximum charge of 4, peptide minimum length of 7 and maximum length of 52 for identification (35 was used as maximum length for quantification). False discovery rate (FDR) of 0.01 for identification of peptide-spectral matches, peptides and protein groups. Quantity MS level – MS2 (base quantity unit was TMT reporter intensities), Minor group (peptide) was quantified by summing using the top 3 method. Major group (protein group) was quantified by summing the top 5 peptides, MaxLFQ option was "on", single hit protiens were not excluded. For the phosphoproteomics dataset (phSEC), raw files were searched similarly with an additional variable phospho(STY) modification, with PTM localization workflow as follows: phosphosite flanking region – 7 amino acids, multiplicity "True", PTM consolidation method – "SUM", phosphosite localization probability filter set to 0.

For DIA runs we used the equivalent parameters in SpectroNaut, with minor modifications: Major group (protein group) was quantified by summing the top 3 peptides, Quantity MS level – MS2, Quantity type – Area, for phUF data an additional variable phospho(STY) modification was defined, PTM localization workflow as follows: phosphosite flanking region – 7 amino acids, multiplicity "True", PTM consolidation method – "SUM", phosphosite localization probability filter set to 0.5. Cross run median normalization and global imputation were used for global expression analysis (HEK-HCT unfractionated samples). Peptide spectral matches (PSMs), peptides and protein group data were exported for subsequent analysis, as well as a PTM site report when applicable.

**PTM localization probability.** To maximize phosphopeptide coverage in phSEC, we did not filter the phosphopeptides based on site localization probability. Our reasoning was that the localization probability does not reflect whether a peptide is phosphorylated. Rather, it reflects the chances that a specific amino-acid out of a few potential sites (Serine, threonine, or Tyrosine) in the sequence is the one phosphorylated.

In addition, we report the number of phosphopeptides measured based on the number of stripped-sequences identified. This means that multiple alternative phosphorylation sites are collapsed and counted as a single phosphopeptide. Notably, although we did not filter phosphopeptides out based on low localization probability, unless otherwise mentioned, all the specific phosphorylation sites that are mentioned in the text were verified to be >0.9 probability.

**Signal processing.** The peptide intensities were spread out along ~55 SEC fractions. Peptides were filtered by being proteotypic and non-decoy. Empty or NA measurements were converted to zeros since: (1) we found it performed best for interaction analyses, in line with prior literature[19], and (2) downstream peak-picking and alignment were unaffected by conversion type due to a minimum noise cutoff. Next, a single Uniprot ID was assigned to each peptide. In TMT experiments, peptide reporter intensity values were normalized to their respective MS1 peak intensity. In experiments conducted with the full overlap

TMT mixing scheme, TMT batch effects were corrected based on the signal in the common fractions between any two adjacent mixes. A normalization factor was calculated by dividing the peptide fraction intensities of mix [n + 1] by mix [n], then taking the median of all the peptides and the mean of all the overlapping fractions in common between the mixes. Mix [n + 1] was then normalized to mix [n] by multiplying all intensities by the normalization factor. Lastly, the peptide intensities of overlapping fractions were averaged.

Then, for the HEK-HCT dataset, the intensities were normalized between conditions. To do this, the median intensities were calculated in each sample (ptm/replicate/condition) and then the medians were used to find an intensity ratio between the samples, which was then used to normalize HCT intensities to HEK intensities. The signal from the two independent biological replicates was then averaged (per protein/peptide in each dataset). Replicate averaging was performed to reduce noise and better model elution peaks for downstream analysis. For plotting and matching purposes, the intensities were smoothed using scipy.signal.filtfilt a linear digital two-way filter (b = [1.0/2]*2, a = 1).

Two averaged and normalized datasets were generated as described above to create smoothed elution profiles. The first consisted of protein-level global intensities and peptide-level phospho-enriched intensities. This dataset was generated to analyze the assembly states on a protein level given the higher amount of protein ID overlap. The second dataset was on the peptide level for both the global and the phospho-enriched intensities. This dataset was used for peptide-level overlap and heatmap generation for comparisons between gSEC and phSEC to more accurately compare intensities between the datasets.

**SECAT.** SECAT was used to identify previously reported protein interactions for the DIA and MX comparison[22,64]. Replicates were analyzed in the same run to leverage the predictive power of the classifier. SECAT analysis was conducted on the processed peptide level signal (as mentioned above) using the default (SECAT provided) positive and negative interaction networks for the training step, and a target database of STRING's human interactions (9606.protein.links.v11.5) for the query step. Additionally, we confirmed the SECAT model's ability to distinguish between the positive and negative interactions in both the SEC-MX and the SEC-DIA data (Supplementary Fig. 2A, B). The default SECAT parameters were set except for a 'pi0_lambda' of 0.4 0 0 0, an 'ss_initial_fdr' of 0.5 and 'ss_iteration_fdr' of 0.2 during the 'learn' step. Additionally, the 'export_tables' option of the SECAT 'learn' step was used to export tables for extracting the STRING target and learning interactions along with their scores. The HEK-HCT data was also quantified by setting HEK as the 'control_condition', and using a 'maximum_interaciton_qvalue' of 0.1 for the quantify and export steps.

The networks were obtained by setting a q-value cutoff of 0.05 on the exported network tables. However, we initially observed that only 34% of the interactions were mutual to both DIA and SEC-MX datasets, but found a large number of interactions exclusive to each dataset at a q-value < 0.05 cutoff were very close to the cutoff in the other experimental setup (Supplementary Fig. 2C). Therefore, we adjusted the cutoff to include any interaction with a q-value between 0.05 and 0.1 (in at least 3 out of 5 SECAT runs), if its q-value was lower than 0.05 in the other dataset. With this adjusted cutoff we observed that 54% of interactions were identified in both datasets, while 29% were only in the DIA dataset and 17% only in the TMT dataset (Fig. 1H).

**EPIC.** The EPIC tool was used to identify high confidence interactions allowing the discovery of interactions independent from a reference database[24]. Replicates were analyzed in the same run to leverage the predictive power of the classifier. Peptides were collapsed to the protein level by adding the top three peptide intensities for each

protein. Proteins that eluted in only one fraction were filtered out. Pairwise protein-protein similarities were then computed using the Pearson Correlation-Coefficent (with and without noise), Jaccard, Apex, Mutual Information, and Euclidean metrics respectively. A cutoff of 0.5 for the features was chosen prior to analysis by a Random Forest Classifier which was trained on reference complexes generated using CORUM, INTact, and GO human proteins. The classifier was trained using an 80/20 cross validation split to minimize variance across runs and maximize predictive capabilities. Finally, de-novo protein-protein interactions were found by querying the classifier and reporting every interaction above 50% confidence as an interaction. To further benchmark the classifier a precision-recall graph was generated by varying the confidence of the classifier and reporting the metrics, the intersection of the precision and recall occurs at 60% confidence. However, as we are trying to minimize false positives, we picked a higher confidence of 80% (as previously reported by Pourhaghighi et al.[25]) which has less interactions with higher precision. Confidence score cutoffs were further adjusted as elaborated for the SECAT dataset, to include any interaction with an EPIC score between 0.6 and 0.8, if its score was higher than 0.8 in the other dataset (Supplementary Fig. 2F).

**Molecular weight (MW) estimation and monomeric fraction cutoff calculation.** Molecular size estimations per SEC fraction were performed using a standard (Biorad 1511901), which was injected and measured onto the SEC column at the start and end of each experimental day. The calibration standard's fractions and log MW were input into sklearn.linear_model.LinearRegression to create a log-linear model and predict the MW of each fraction (Supplementary Fig. 8A). In this regard, the monomer fractions were predicted by using the Uniprot determined MW to get an estimated fraction of elution for the monomer. Additionally, a MW multiplier of 1.5 was used to account for wide or slightly shifted elution peaks.

**Peak picking and alignment.** Using the smoothed and averaged elution profiles (as describe in *Signal Processing*), we developed an analysis pipeline called SPADE-PTM (SEC Protein Assembly Dynamics Evaluator for PTMs) to identify and compare assembly states as elution peaks (see Code Availability). The elution peaks were identified using scipy.signal.find_peaks (prominence=max(intensity)*0.05, distance=3, Height=1000) to get the apex position (the identity of the peak) and scipy.signal.peak_widths to get other attributes such as the peak height. In downstream analysis, peaks are synonymous with apex position. Each protein/peptide was expanded into a list of one or more peaks. When no peaks were identified, the protein/peptide was dropped due to noise. Of note, only one protein out of the 1596 proteins overlapping between gSEC and phSEC was removed due to the inability to model any peaks.

With peaks identified in each sample, we then matched them within protein/peptide (protein-centric) between PTM enrichment and conditions. By mapping a phosphopeptide to a global peak, the assembly states maintained the same peak apex location as the gSEC peak – allowing many phosphopeptides to map to a single global protein peak. Phosphopeptide (phSEC) peaks were mapped to global (gSEC) peaks with a maximum cutoff of 3 fractions. A threshold of 3 fractions was chosen by comparing peak differences of IDs between both replicates to random pairs – whereby extending the threshold further only showed additional value up to 3 fractions over the decoys (Supplementary Fig. 8B).

Peaks were matched within protein/peptide between condition by finding the best alignment path (described below). The main difference between condition alignment and the PTM mapping is that the assembly state's peak apex location was calculated as an average of the aligned peaks. To do this, we first started with the list of identified peaks for a given protein/peptide in a sample (condition/PTM) as

follows, where $L_i$ is the list of peaks for the $i^{th}$ sample, out of $n$ samples. Furthermore, $L_{total}$ represents all the peaks of a protein/peptide in all $n$ samples as shown in Eq. 1.

$$L_{total} = [L_1, L_2, \ldots, L_n] \tag{1}$$

Then, to account for non-matching peaks between samples (i.e., the peaks are too far away and therefore are not identified in one or more sample), we added a NaN value to each list of peaks ($L_i$). The operation was performed where $l_{ij}$ are the individual peaks of $L_i$, and $j$ is a given peak out of the $k$ peaks in sample $i$ (Eq. 2).

$$L_i' = [l_{i1}, l_{i2}, \ldots, l_{ik}, NaN] \tag{2}$$

Next, we aligned the peaks between $n$ samples of a protein/peptide to create possible alignment paths, such that a single path $x$ is defined in Eq. 3. Each possible path of a given protein will differ depending on the location of the peak or whether a NaN was substituted (indicated by $j$).

$$Path_x = \left\{ l_{1j}, \ldots, l_{nj} \right\} \tag{3}$$

To find all possible combination, we considered every possible path between the $n$ samples and their $k$ peaks. In other words, each $Path_x$ is comprised of $n$ elements, one element for each sample (with each element as either a peak or a NaN). The total number of paths scored per protein/peptide can be calculated as the product of the length of each peak list as shown in Eq. 4.

$$All\ Paths\ (per\ protein\ or\ peptide) = \prod_{i=1}^{n} |L_i'| \tag{4}$$

With all possible peak paths, we wanted to find the best non-redundant paths to determine the optimal peak alignment. To do this, we first needed to score each path. First, we found the mean peak location of each path as shown in Eq. 5, where $m$ is the number of non-NaN peaks in a $Path_x$ and $l_{ij}$ peaks represented by NaN were substituted with zeroes.

$$mean_{path_x} = \frac{1}{m} \sum_{i=1}^{n} \begin{cases} l_{ij} \text{ if } l_{ij} \text{ is not NaN} \\ 0 \text{ if } l_{ij} \text{ is NaN} \end{cases} \tag{5}$$

Then variance of the paths the paths were calculated as shown in Eq. 6, where the NaN peak values were substituted for the Threshold input. In other words, peaks further away from the mean than the Threshold would give a higher variance than substitution for a NaN.

$$var_{path_x} = \frac{1}{n} \sum_{i=1}^{n} \begin{cases} \left( l_{ij} - mean_{path_x} \right)^2 \text{ if } peak_i \text{ is not NaN} \\ (Threshold)^2 \text{ if } peak_i \text{ is NaN} \end{cases} \tag{6}$$

The path variances were then converted into a score as shown in Eq. 7, where the best scoring path was represented by the lowest variance.

$$Path_x Score = \frac{1}{var_{path_x} + 1} \tag{7}$$

With all the paths scored for a given protein/peptide, we used an iterative method to select the best scoring $Path_x$ ($Path_{best}$). Then, we removed any other $Path_x$ that contained peaks from $Path_{best}$ to avoid redundant alignment paths. Once the non-redundant paths were removed, the lowest scoring $Path_x$ was once again chosen and redundant paths once again removed. This operation was iterated until all peaks ($L_{total}$) were exhausted for a given protein/peptide, whether aligned with other peaks or by themselves.

Using the above formulas, the peak-level data was aligned for each common protein between HEK293 and HCT116 as well as gSEC and phSEC. Fold changes were calculated as log2(HCT116/HEK293) for each assembly state based on identified peak height. For proteins with multiple phosphorylated peptides a peak ratio was calculated per peptide. Additionally, peaks/assembly-states were classified as in the complex or monomer region based on the mean peak apex position of the matched peaks. Further details on the peak-picking and alignment can be found and reproduced on the GitHub page for SPADE-PTM.

**Quantitative differentials.** The SEC-MX fold change ratios (HCT116/HEK293) were calculated on an aligned peak-level (as described above) using the peak heights of the smoothed and replicate-averaged signal (per protein assembly state). Phosphopeptide peaks were calculated separately for each phosphopeptide that mapped to the same gSEC peak in both cell lines. In other words, some phSEC fold-changes were lost due to not aligning to the gSEC peak in one of the cell lines. The unfractionated (UF) ratios were calculated using the replicate-averaged intensity values per protein ($n = 3$).

**Enrichment analysis.** Response groups (Fig. 7A) were determined based on a cutoff of +/- 2 fold-change in either gSEC or phSEC. Nine response groups were determined based on the combination between the 3 possible enrichments in each dataset (HEK293, HCT116, or Not Significantly different). Enrichment analysis was then conducted on the proteins represented by the peaks in each response group. Enrichment analysis was performed using the WebGetalt [http://www.webgestalt.org] platform using the over-representation analysis (ORA) against the Reactome Pathway[74,75] database. Kinase target over representation analysis was conducted similarly on the same platform. Enriched sets were compared to a background list containing all the proteins identified in the nine response groups.

**Reporting summary**
Further information on research design is available in the Nature Portfolio Reporting Summary linked to this article.

## Data availability
The mass spectrometry data generated in this study have been deposited in the MassIVE database. Data for HEK293 SEC-DIA (used in the DIA-MX comparison as well as in the comparison of +/- RNase treatment) under accession code MSV000093915 [https://massive.ucsd.edu/ProteoSAFe/dataset.jsp?task=66dc6dc681d54a56b030fcd95fbdb975]. The HEK293 SEC-DIA can be downloaded via the FTP download link [ftp://massive.ucsd.edu/v06/MSV000093915/]. All other data (HEK293 and HCT116 SEC-MX and unfractionated samples) under accession code MSV000096001 [https://massive.ucsd.edu/ProteoSAFe/dataset.jsp?task=9afc199494e74f4ebafb8f45a522b548] and can be downloaded via the FTP link [ftp://MSV000096001@massive.ucsd.edu]. The data are also associated with a ProteomeXchange deposition under accession code PXD059734. This includes LC-MS/MS raw files, search engine files (.psar from Spectromine for all TMT-based searches and.sne from Spectronaut for all DIA-based searches) and reports in tabular format, as well as normalized values used in downstream analyses. The normalized values (after signal processing) used in downstream analyses are also available as AWS downloadable links. Input tables: [https://sec-mx-example-data.s3.amazonaws.com/data_inputs.zip], output tables: [https://sec-mx-example-data.s3.amazonaws.com/data_outputs.zip]. In addition, the processed gSEC and phSEC data can be found in Supplementary Data 1 and 2, respectively. The list of proteins from Fig. 5D (peaks found exclusively in each cell-line's dataset or mutual to both) can be found in Supplementary Data 3. The unfractionated data (gUF and phUF) can be found in Supplementary Data 4. The list of proteins

from Fig. 7 (proteins with peaks falling in each one of the 9 response groups defined by the scatterplot in Fig. 7A) can be found in Supplementary Data 5. Unless otherwise stated, all data supporting the results of this study can be found in the article, supplementary, and source data files. Source Data are provided with this paper.

## Code availability

Scripts and source data used in this study are available from GitHub for SPADE-PTM: [https://github.com/mjlab-Columbia/SPADE-PTM][76], [https://doi.org/10.5281/zenodo.14219108].

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

## Acknowledgements

BJB was supported by NSF-GRFP (Award DGE2036197). LCT was supported by NSF-GRFP (Award DGE2036197). ED-M was supported by NIH/NINDS (K99NS135103). NHS was supported by NIH/NIGMS (R35GM138014). MJ was supported by NIH/NIA (R01AG071869). We thank the Yeo lab (UC San Diego) for kindly gifting us HEK293XT cells, and the Prives lab (Columbia University) for their kind gift of HCT116 cells. We thank Cassandra A. Chartier for her assistance with HPLC.

## Author contributions

ED-M, BJB, and MJ designed the study. ED-M, BJB, YM, LAS, and LCT carried out the experimental work: ED-M, YM, and LAS fractionated the samples, ED-M, YM, and LCT conducted labeling and phosphopeptide enrichment, ED-M, BJB and YM processed the samples for mass spectrometry. BJB developed the software. ED-M, YM, BJB, GR, LAS, and AAA analyzed data and visualized results. ED-M, YM, BJB, and MJ wrote the manuscript draft, all authors revised and approved the final version. MJ acquired the funding. ED-M, MJ, and NHS supervised the study.

## Competing interests

The authors declare no competing interests.
