## [Transparent Peer Review file · Nature Communications]

SEC-MX: An approach to Systematically Study the Interplay Between Protein Assembly States and Phosphorylation

Corresponding Author: Professor Marko Jovanovic

Parts of this Peer Review File have been redacted as indicated to remove editor comments.

Version 0:

Reviewer comments:

Reviewer #1

(Remarks to the Author)
General Comment

The authors propose the use of SEC coupled with PTM enrichment to detect PTM-specific protein assembly states. Overall, the manuscript presents an original technological breakthrough with significant potential for studying biological states. The manuscript is well-written and follows a coherent order. Key components of the proposed technology, such as the full-overlap scheme that allows for technical duplicates and the multiplexed design for measuring different biological samples, are well-highlighted. This manuscript is an excellent example of leveraging the increased multiplexing capacity of modern TMT labeling to study phosphorylation-dependent complexes.

Major Points

1. Resolution and Aggregated Signal: The manuscript does not address what this new level of resolution provides that cannot be captured by an aggregated signal. For instance, if the number of observations per biological condition (cell line in this case) is large enough, what can be learned from the SEC component that cannot be learned by mapping differential peptide abundance changes into a generic PPI network? To address this, the authors could exemplify how, for some PTM-specific differential assembly states, the corresponding aggregated signal (by summing along the axis of fractions) does not show any changes between biological contexts. This analysis is currently lacking and constitutes the baseline for the proposed technology.
2. Enrichment Efficiency and Computational Details: What is the enrichment efficiency of the IMAC enrichment? Specifically, how many phosphopeptides and non-phosphopeptides are detected in the phSEC fractions? Additionally, fundamental details about the computational processing of the phosphoproteomics component are missing. What PTM localization probability strategy is used? What is the cutoff employed? Further details are needed.
3. Biological Relevance: While comparing two cell lines can be interesting, it does not represent an ideal example where phosphorylation is expected to play a fundamental role in protein assembly states. To showcase the potential of this technology, an experiment involving a stimulus expected to produce significant phosphorylation changes in the same biological system should be used. This is particularly relevant in short-time points when most signaling events occur. Examples could include DNA damage or stimuli that trigger phosphorylation cascades, such as growth factors.

Minor Points

1. Figure 1C Details: The sentence "We found that coverage in SEC-MX was comparable to SEC-DIA, with only a 10% difference in protein-group identifications, despite a 9-fold reduction in the number of LC-MS/MS runs (57 versus 8 in DIA or TMT, respectively) (Figure 1C)" could be improved by making the number of LC-MS/MS runs explicit in Figure 1C.
2. Cost Comparison: While the technology is compared with DIA-based SEC, showing increased throughput per LC-MS/MS

run, it is important to mention the cost of TMT labels in the discussion. Including a small (not necessarily accurate) comparison of the expenses needed to perform each method would be beneficial.

(Remarks on code availability)

Reviewer #2

(Remarks to the Author)

The authors proposed SECMX (Size Exclusion Chromatography fractions MultipleXed), a global quantitative method combining Size Exclusion Chromatography and PTM-enrichment for simultaneous characterization of PTMs and assembly-states. They generated a proof-of-concept dataset in HEK293 and HCT116 cells mapping thousands of phosphopeptides and their assembly-states. There are several problems:

1. The authors should provide the dataset of HEK293 and HCT116 cells.
2. What is the relationship between https://github.com/mjlab-Columbia/peak_alignment and SECMX (Size Exclusion Chromatography fractions MultipleXed)? Is pyPA developed for SECMX? There are only few simple scripts. The authors should provide all the data and codes used in this study in GitHub as a resource.
3. In Figure 1 – SEC-MX performs comparably to SEC-DIA in coverage and resolution, F, what was the y-axis? In Figure 1 H and Extended Data Figure 4 – gSEC and phSEC datasets comparisons (A) Venn diagram, what was the background and what was the overlap p value?
4. In Figure 2 – SEC-MX enables phosphopeptide enrichment C, the four figures were very similar? Could the authors highlight the different parts, or evaluate it quantitatively?
5. In Figure 3 – A novel protein-centric analysis framework to study assemblystates, what was condition 1 and condition 2? The authors should add the condition names rather than numbers?
6. In Extended Data Figure 3 – SEC-MX performs on par with published methods, at an order-of magnitude reduction in LC-MS/MS runs, the number of high-confidence interactions for SECMX was very low. Can these methods be ensembled to increase both the number high-confidence interactions and the number of PPIs per MS run?
7. In Figure 6 – a peak-centric analysis uncovers different regulation of distinct assembly-states and Extended Data Figure 6 – HEK-HCT Differential and Unfractionated Comparison, E and F, there were only “-” and up of Global? Were there down functions?

(Remarks on code availability)

What is the relationship between https://github.com/mjlab-Columbia/peak_alignment and SECMX (Size Exclusion Chromatography fractions MultipleXed)? Is pyPA developed for SECMX? There are only few simple scripts. The authors should provide all the data and codes used in this study in GitHub as a resource.

Reviewer #3

(Remarks to the Author)

Summary:

The authors present a novel approach using Size Exclusion Chromatography (SEC) coupled with Tandem Mass Tagging (TMT) and mass spectrometry to analyze protein-protein interactions (PPIs) and post-translational modifications (PTMs) in HEK293 and HCT116 cells. The integration of SEC with TMT labeling enables high-throughput, quantitative analysis, identifying numerous PPIs and providing insights into protein assembly states and their regulation by PTMs. The authors combined mass spec data with a SECAT analysis that effectively predicts and validates PPIs, constructing high-confidence interaction networks and finally performed an EPIC analysis, that adds robustness by integrating functional association data and complement the SECAT analysis. Additionally, the SEC-MX method was compared to a DIA workflow without labeling and multiplexing. The mass spec measurements have been carried out on a Q Exactive HF instrument.

The study's findings are significant for advancing the understanding of protein interactions and regulation of complex assemblies through PTMs (here phosphorylation) and allows for comprehensive mapping of both together, providing a dual perspective that is often challenging to achieve. The insights gained from the study could advance the understanding of protein assembly states and their regulation, potentially informing research in cellular mechanisms and disease states.

General Remarks:

The data obtained from SEC-MS is technically sound and presented in sufficient detail. However, the manuscript would benefit from addressing issues regarding ratio compression in TMT quantification as well as normalization of phosphorylation sites to a reference proteome, providing more details on the machine learning classifier's validation for SECAT, and discussing false discovery rates in peptide and protein identifications. Additionally, the authors state that the new method saves MS measurement time due to the multiplexing approach and can hence result in more protein-protein interactions per MS run but it is not discussed that the sample preparation procedure of fractionation plus TMT is very time-consuming and the complex experimental design is challenging to establish especially for labs that do not commonly perform TMT experiments. Please find a detailed point-to-point list below:

Minor Points:

- The manuscript would highly benefit from an overview figure that describes what kind of samples have been processed with what kind of workflow and analyzed with what kind of software pipeline and additional software packages. It is partially done in Figure 1 and 2 but only until MS measurements without data analysis pipeline used afterwards. A full overview of all experiments, pipelines and so one would help the reader to get an easy entrance point

Figure 1:

- Maybe this is just due to zooming in but why does the profile comparison between DIA and MX in (E) look different compared to the full profiles in D?
- (F) it is unclear what is plotted on the y-axis
- (G) would be good to clarify that these numbers are from the SECAT analysis directly in the figure (headers maybe)

Figure 2:

- (A) it is not clear from the figure whether the IMAC enrichment was performed on the total TMT pool or on the TMT pool of cells but for each of the 54 fractions separately. This information is also missing in the method section.

Extended Data Figure 3:

- The reference data sets for performance comparison are quite outdated, for example the crosslinking data set is from 2021, Bartolec et al. (2023) (<https://doi.org/10.1073/pnas.2219418120>) reported 28,910 unique residue pairs, 4,084 unique proteins, and 2,110 unique protein-protein interactions from a crosslinking experiment coupled with SEC. (approximately 482 interactions per MS run)
- Taking interactions per MS run can be quite misleading because it does not consider the MS run time. One could choose longer gradients to alter the position of the study in the diagram. In fact, the total run time of SEC-DIA in this manuscript is 70min, while DIA (unfractionated) gradients are 160min and SEC-MX gradients are 150min. Was the total run time somehow normalized before plotting it? It is not stated in the figure legend

Extended Data Figure 4:

- (A) Is the peptide intersection of HEK and HCT cells together or just one cell type?
- (B) If the overlap between gSEC and phSEC is so little how can the profiles between both methods be so similar? If just these 960 overlapped peptides are plotted here, then this must be stated somewhere, at least in the figure legend

Method section:

- The FDR settings used for searches must be mentioned in the data analysis section, but this is not the case, please add this information
- Although it is stated that the BSG factory settings are used it is good practice to write at least some details about search settings that have been used for example: modifications used, ppm error on MS1 or MS2, was the quantification performed on MS1 or MS2 level? And so on. Spectronaut 16 is quite old now and factory settings certainly have changed.
- Conversion of empty or NA values to zero is one method that can be used but also not the best for this kind of analysis. Have other techniques been evaluated for example Nearest neighbor imputation?
- Please provide detailed information on the training and validation of the machine learning classifier used in SECAT analysis. Include performance metrics such as precision, recall, and F1 scores.
- As far as I get it 2.5 µg peptides were injected for each MS run. I think this is quite a lot, especially for 70 min gradient run. Was the carry over evaluated between the fractions?

Major Points:

- Quantitation of TMT experiments can be tricky, the issue of ratio compression, commonly observed in TMT-based quantification, is not thoroughly addressed in the method section. Ratio compression can lead to inaccurate quantification of peptide abundances, affecting the interpretation of interaction strengths and protein assembly states.
- Additionally, the accuracy of the TMT quantitation can be enhanced by normalization of the phospho-peptide intensities (areas) to a reference proteome that was measured before pooling the fractions. Maybe this was already done, and I have overlooked it but it would be good to have a detailed description in the method section. If this was not performed, there would be a bias in the conclusions about assembly state regulation by phosphorylation towards protein abundance differences instead of phosphorylation differences
- What kind of control was used for protein SEC to test whether complexes are unaffected by the procedure itself? O'Reilly et al 2023 (<https://doi.org/10.15252/msb.202311544>) for example could show that some protein complexes dissociate during the fractionation process but can be stabilized using crosslinking reagents. If this is also the case for protein complex presented in this study the conclusion about difference between phosphorylation states of monomers and full complexes would be biased.

Supplementary data:

- As supplementary data a list of protein names (gene names) was provided but there is no title nor legend or description available. I think this should be added directly in the document
- Data availability: I could not find the .sne files from Spectronaut. It would be nice to provide the full Spectronaut and SpectroMine analysis files

(Remarks on code availability)

Version 1:

Reviewer comments:

Reviewer #1

(Remarks to the Author)

The authors have successfully addressed all of my comments, with the exception of the third major point. However, I recognize the significant improvements in both the quality and content of the revised manuscript, and I therefore recommend

it for publication in its current form. It would be interesting if, in future work, the authors could explore whether they can capture phosphorylation-dependant complex reassemblies following rapid stimulation rather than focusing solely on basal states.

(Remarks on code availability)

I've reviewed the code and appreciate its overall structure and readability. However, I'm unable to access the example input dataset that the repository points to. Below is my feedback so far:

The code is well-documented, and the README provides clear guidance on the various functionalities the Python package offers.

Each of the four main functionalities described in the repository is implemented in its own Python script, with the necessary classes and methods for each analysis step.

Installation of the package environment and dependencies completes without any errors.

Unfortunately, the example data is currently unavailable, as the S3 file cannot be downloaded due to access denial at this time.

Since the data is inaccessible, I'm unable to test the code's functionality at this moment.

Despite this, the code is well-structured, clearly documented, and highly readable.

P.S: This is the link in the repo I can't access.

https://sec-mx-example-data.s3.amazonaws.com/data_input.zip

https://sec-mx-example-data.s3.amazonaws.com/data_input.zip

Reviewer #2

(Remarks to the Author)

I have no further questions.

(Remarks on code availability)

The code looks correct and for each step, there are explanations. But I do not use macOS and have not actually run the code.

Reviewer #3

(Remarks to the Author)

Reviewer comments 2nd round:

The authors have addressed all comments that have been raised. The revised and additional figures improved the understanding and clarity of the method design and results.

I am still concerned about the 69% ratio compression and labelling efficiency of below 95% for two of the TMT mixes but the authors openly reported this issue using figures and discussed it partially in the main text. I appreciate the openness at this point. If the other reviewers are not concerned, I am fine with it like it is because it is mentioned that this is a proof-of-concept study and there is room for improvement.

The missing details in the method sections have been addressed fully and two Spectronaut .sne (5 .psar) files have been added to the repository.

The manuscript was restructured and clearly shows the complexity of the method and the effort that needs to be made to implement this method. On the other hand, the biological insights are highlighted as well with additional figures focusing on protein pathways with integrated assembly-state arising from their study for example.

(Remarks on code availability)

A Multiplexed SEC-MS Approach to Systematically Study the Interplay Between Protein Assembly-States and Phosphorylation Events

We sincerely thank the reviewers for their thorough consideration of our manuscript and their insightful comments. We are pleased to submit a revised version that we believe has been significantly improved based on this feedback. Among the key revisions, we completely rewrote the sections focused on the differential analysis between HEK293 and HCT116 samples, including Figures 5-7 and Extended Data Figures 5-7. The updated sections now emphasize the biological relevance of our findings, with more detailed discussions on the choice of biological systems and the selection of specific examples. Additionally, we expanded on the advantages of SEC-MX and its ability to resolve assembly states by comparing these results with bulk measurements of unfractionated samples. This analysis is now included in a dedicated Figure 6 and features a new dataset of phosphopeptides enriched from unfractionated samples.

We also incorporated previously missing technical details, such as the specifics of phosphopeptide enrichment (IMAC efficiency, PTM localization filters), TMT multiplexing, and more comprehensive figures to help readers better understand the experimental pipelines. Furthermore, we updated our data and code repositories to address availability concerns. We believe that the manuscript is now much improved thanks to the reviewers' valuable feedback, and we hope that it will provide clearer insights for our readers.

Point by Point answers to the reviewers' comments

Original reviewer comments are marked in black / bold, authors response is written in blue.

Reviewer #1 (Remarks to the Author):

General Comment

The authors propose the use of SEC coupled with PTM enrichment to detect PTM-specific protein assembly states. Overall, the manuscript presents an original technological breakthrough with significant potential for studying biological states. The manuscript is well-written and follows a coherent order. Key components of the proposed technology, such as the full-overlap scheme that allows for technical duplicates and the multiplexed design for measuring different biological samples, are well-highlighted. This manuscript is an excellent example of leveraging the increased multiplexing capacity of modern TMT labeling to study phosphorylation-dependent complexes.

We thank the reviewer for their thorough read and thoughtful consideration of our work.

Major Points

1. Resolution and Aggregated Signal: The manuscript does not address what this new level of resolution provides that cannot be captured by an aggregated signal. For instance, if the number of observations per biological condition (cell line in this case) is large enough, what can be learned from the SEC component that cannot be learned by mapping differential peptide abundance changes into a generic PPI network? To address this, the authors could exemplify how, for some PTM-specific differential assembly states, the corresponding aggregated signal (by summing along the axis of fractions) does not show any changes between biological contexts. This analysis is currently lacking and constitutes the baseline for the proposed technology.

We appreciate the reviewer's insightful comments and we agree that including comparison(s) between SEC results and corresponding quantifications of aggregated signal is a major part of the proof-of-concept, which was not highlighted enough in the original manuscript.

To address this, we have conducted additional analyses on the "global" (not enriched) unfractionated (gUF) samples, as well as conducted additional measurements of phosphopeptides enriched from unfractionated samples (phUF). We now dedicate two full figures for these comparisons between SEC and UF (Figure 6, Extended Data Figure 6). This is illustrated through scatterplots comparing the UF and SEC differential quantification data (Figure 6A, Extended Data Figure 6A-C). We further discuss how comparing differential HCT116/HEK293 ratios in UF to ratios based on distinct SEC peaks (Figure 6A) reveals a wider distribution of fold changes compared to the equivalent UF versus SEC comparison that is based on averaging all the peaks of a given protein (Extended Data Figure 6A), or to a comparison based on summing signal intensities across all fractions in SEC (Extended Data Figure 6B). This is evident by the change in R values from 0.55 (individual peaks), to 0.63 (averaged peaks), to 0.72 (summed intensities), which is elaborated in the figure legend of Extended Data Figure 6 (page 13 and 46).

In addition, we highlight two examples: CDK4 (Figure 6B-C) and RPTOR (Figure 6D-G), showing how the inclusion of the SEC dimension allows us to distinguish between assembly states that have distinct differential ratios — something that cannot be captured by the aggregated signal. To further clarify this point, we include unfractionated (abundance-level) phUF and gUF bar plots for CDK4 and RPTOR, providing a quantitative comparison to the aggregated signal.

In addition, in order to address how SEC contributes added value compared to overlaying abundances on a PPI network we compared our SEC data to the BioPlex interaction network (Huttlin et al., 2021). One such example is CDC37 (Figure 5E-F), which shows a difference between HEK293 and HCT116 in its distribution across two assembly-states. This difference is not evident in the BioPlex interaction network for CDC37 (Extended Data Figure 5E), highlighting how the assembly-state resolution enabled by SEC allows for additional information over affinity-purification mass-spectrometry (page 12).

Similarly, we show how the abundance ratios measured by SEC contribute to confirming the BioPlex authors' hypothesis regarding the GINS-MCM-CDC45 complex (Extended Data Figure 5C-D). While they observed a HEK293-HCT116 differential pattern of interactions among the subunits of the complex in AP-MS, they suggested that this can either reflect rewiring of interactions, or alternatively stem from abundance changes of complex members. Our observed SEC patterns for this complex (Extended Data Figure 5C-D) support the latter hypothesis: while

there are no observed changes in assembly-states between the cell-lines, all the GINS subunits (GINS1-4) show increased abundance in HEK293 compared to HCT116 cells, highlighting the quantitative advantage of SEC-MX (page 45).

Taken together, we use both systematic analyses and specific examples to demonstrate how SEC provides assembly-state resolution, offering a clear advantage over both bulk measurements and AP-MS networks. We thank the reviewer for highlighting that this aspect was not sufficiently emphasized in the original manuscript, as we now believe these additional analyses form a critical part of the proof-of-concept for SEC-MX.

2. Enrichment Efficiency and Computational Details: What is the enrichment efficiency of the IMAC enrichment? Specifically, how many phosphopeptides and non-phosphopeptides are detected in the phSEC fractions? Additionally, fundamental details about the computational processing of the phosphoproteomics component are missing. What PTM localization probability strategy is used? What is the cutoff employed? Further details are needed.

We apologize for not adding sufficient information about the enrichment efficiency and computational details of our phosphoproteomics analysis. To address these concerns, we have added the requested information on IMAC enrichment efficiency in Extended Data Figure 3A, and reported the efficiency of 96.6 +/- 2.4% (mean +/- stdev) in the Materials and Methods section (page 24-25). Additionally, we have clarified the workflow used for searching and analyzing the phosphoproteomics data, also included in the Materials and Methods section (page 25-26).

Specifically, we detail our choice not to use a probability localization filter in exporting the phosphopeptide data. We explain that this choice was done in order to avoid potential loss of phosphopeptide-coverage. This decision is based on the reasoning that "localization probability" reflects the probability of which site is phosphorylated (among a few alternative optional S/T/Y in the sequence), rather than determining whether the peptide itself is phosphorylated or not. Therefore, we did not want to exclude peptides from the analysis on the basis of uncertainty about which site is phosphorylated. However, if specific phosphorylation sites are mentioned in the manuscript, the localization was verified with a probability over 0.9. Lastly, in our downstream analyses, we report the number of phosphopeptides based on unique stripped sequences, without considering multiplicity or alternative phosphorylation sites on the same peptide, and therefore multiple alternative sites are only counted once towards the number of phosphopeptides in the analysis.

We hope this additional information in the manuscript helps clarify our enrichment approach both to the reviewer and to the readers, as well as addresses the reviewer's concerns.

3. Biological Relevance: While comparing two cell lines can be interesting, it does not represent an ideal example where phosphorylation is expected to play a fundamental role in protein assembly states. To showcase the potential of this technology, an experiment involving a stimulus expected to produce significant phosphorylation changes in the same biological system should be used. This is particularly relevant in short-time points when most signaling events occur. Examples could include DNA damage or stimuli that trigger phosphorylation cascades, such as growth factors.

We fully agree with the reviewer that an important aspect of validating SEC-MX is showing its value in a biological system with expected changes in phosphorylation that contribute to assembly-state changes. Therefore, we thank the reviewer for pointing-out that this aspect was not adequately addressed in the previous submission of the manuscript. However, we do believe that the HEK293 versus HCT116 comparison is well positioned to show the full potential of SEC-MX for the following reasons:

1. HEK293 and HCT116 have known differences in the activation of signaling pathways, resulting from their respective driver modifications to generate the cell-lines. HCT116 is a cancer cell-line, while HEK293 was immortalized by adenovirus transformation – resulting in expected differences in signaling pathways which affect both phosphorylation and assembly-states, such as P53 and MTOR signaling (elaborated below).
2. Thanks to a recent seminal work in the field (Huttlin et al., 2021 – “BioPlex”), the HEK293 versus HCT116 is currently the deepest comparative dataset on protein-protein interaction differences between two biological systems. Importantly, this dataset was generated using a method orthogonal to SEC-MS, providing a fully independent reference dataset on protein-protein interactions.
3. In addition, this orthogonality is of great value to the PPI research community as it provides two global comparative datasets (AP-MS and SEC-MS) each with its own strengths and caveats, which are highly complementary and can be systematically evaluated in future studies. Therefore, having a high-quality complementary SEC-MS dataset profiling HEK293 versus HCT116 provides a valuable resource to the community.

However, as pointed out by the reviewer, these advantages of our dataset and comparisons were glossed over and not properly addressed in the prior submission. To address this problem, we have made extensive necessary revisions and reconstructed 3 full figures (Figure 5-7, and their Extended Data Figures 5-7) highlighting the expected differences between HEK293 and HCT116 as described in BioPlex and other works (Page 11-13: “SEC-MX enables differential comparison between biological samples”). Together, this constitutes a substantial part of the results section, which is now revised in order to address this critical issue in a more appropriate manner.

Specifically, the following changes were made to address the issue of expected biological differences:

- We now dedicate a section to describing and justifying the choice to investigate HEK293 versus HCT116 cell-lines, highlighting their biological differences (Page 12, Figure 5A).
- We make direct comparisons between SEC-MX results and main observations in BioPlex, which were not included in the previous version of the manuscript (Figure 5 and Extended

Data Figure 5). This is shown with the GINS-MCM-CDC45 complex, COP9 Signalosome, and Ribonuclease P. These comparisons are used for validation, as well as highlight SEC-MX's unique advantages as a quantitative tool to study assembly-states with sub-complex resolution. For example, we show how SEC-MX was able to confirm a hypothesis made by the BioPlex authors, that the differential interaction pattern observed for the GINS complex seems to be mainly driven by abundance differences, rather than differences in the co-elution patterns of the subunits (Page 11 and Extended Data Figure 5C-D).

- We mindfully selected the specific examples that are brought in various sections of the revised manuscript that are dedicated to the differential analysis (Figure 5-7). Our choices now highlight cases of biological differences that were previously reported or expected between HEK293 and HCT116: CDC37, CDK4, RPTOR, and P53.
- We conducted unbiased enrichment analyses on the proteins that were observed with differential assembly-states (figure 7A-B, Extended Data Figure 7A-B). We delineated pathways that were previously reported to be differentially activated in HEK293 and HCT116: Transcriptional regulation by TP53 and MTORC1-mediated signaling. Signaling downstream from p53 is known to be activated in HEK293 cells due to their original adenovirus transformation, while the mTOR pathway is expected to be activated in HCT116 cells due to their KRAS mutation that leads to activation of both the RAF/RAS/MAPK and PI3K/AKT/mTOR pathway. Since the results of this undirected analysis are in-line with previous literature, it supports the use of HEK293 and HCT116 to explore biological differences (Page 14-15).
- We curated a signaling pathway based on these pathways (p53 and mTOR signaling), annotating SEC-MX observations about quantitative differences in phosphorylation as well as assembly-state abundances, showcasing the rich information contents enabled by the method (Figure 7C), and that it enables investigation into signaling cascades.

We would like to thank the reviewer again for insightfully bringing up this essential point since we believe these revisions enhance the paper. Based on the reviewer feedback, we put a greater emphasis on the biological differences between the cell-lines and focused the paper on known and expected differences in phosphorylation and assembly-states – which demonstrate the capability of the method. Overall, we hope this addresses prior concerns about the biological application of SEC-MX.

Minor Points

1. Figure 1C Details: The sentence "We found that coverage in SEC-MX was comparable to SEC-DIA, with only a 10% difference in protein-group identifications, despite a 9-fold reduction in the number of LC-MS/MS runs (57 versus 8 in DIA or TMT, respectively) (Figure 1C)" could be improved by making the number of LC-MS/MS runs explicit in Figure 1C.

We followed the reviewer's suggestion and clarified the number of LC-MS/MS runs in Figure 1C. In response, we have updated the figure to explicitly include the number of runs for each method. To further enhance the clarity of the comparison between the methods, we have also included

the instrument time required for each method. We hope these modifications provide clearer insights for the readers and address the reviewer's concerns effectively.

2. Cost Comparison: While the technology is compared with DIA-based SEC, showing increased throughput per LC-MS/MS run, it is important to mention the cost of TMT labels in the discussion. Including a small (not necessarily accurate) comparison of the expenses needed to perform each method would be beneficial.

We agree that the cost of TMT labeling is significant and should be a key consideration for researchers when choosing to conduct SEC-MX versus SEC-DIA, in addition to the significant additive "hands-on" experimental workload. We estimated the overall costs as elaborated below, and reached the conclusion that the costs are fairly similar:

- We take into account that an average LC-MS/MS run with a short gradient costs ~\$100 at a typical mass spectrometry core facility. Therefore, a single replicate of SEC-DIA with 72 fractions will cost ~\$7,200 to run.
- With our labeling protocol, a full SEC run (72 fractions) with the full overlap scheme requires ~50% of a 10 Unit TMTpro18 kit, which is ~\$5,000.
- We consider that a long gradient LC-MS/MS run, as required for TMT runs, costs ~\$200 at a typical mass spectrometry core facility. In addition, each mix requires an additional short gradient run (\$100) to verify labeling efficiency – together, each TMT run is estimated at \$300, for a total of \$2,400 for 8 TMT mixes.
- Together with the cost of reagent it sums up to ~\$7,400 for the TMT-based SEC, which is about equivalent to the formerly calculated \$7,200 for a SEC-DIA experiment.

We decided not to include these cost figures, as we felt there were many confounding factors involved in the calculation (such as access to mass spectrometry, etc.). However, we now mention the additional cost associated with TMT runs, as well as the extra time and expertise this experimental design requires, in the discussion section (Page 20).

In addition, considering this comment, as well as others made by the reviewers, we decided to be more subtle in presenting SEC-MX's advantages in increasing throughput. We now emphasize in the discussion that in our opinion the primary advantage of SEC-MX lies in its ability to enable phosphopeptide enrichment as well as precise quantitative comparisons between assembly and phosphorylation states, rather than in the increased throughput. Given this focus, we also opted to remove the figure comparing [PPIs per MS run] in published works (previously Extended Data Figure 3), as we did not feel it provided a fair assessment of different methods when based solely on the number of interactions and mass spectrometry runs, which involves factors like cost of reagents, experimental workload and instrument time.

We hope that these revisions help address the reviewer's concerns and provide clarity for readers regarding the advantages and limitations of the method.

Reviewer #2 (Remarks to the Author):

The authors proposed SECMX (Size Exclusion Chromatography fractions MultipleXed), a global quantitative method combining Size Exclusion Chromatography and PTM-enrichment for simultaneous characterization of PTMs and assembly-states. They generated a proof-of-concept dataset in HEK293 and HCT116 cells mapping thousands of phosphopeptides and their assembly-states.

There are several problems:

1. The authors should provide the dataset of HEK293 and HCT116 cells.

We apologize for the misunderstanding and for any confusion regarding the availability of the data. We had intended to provide the dataset upon submission and have now double-checked to ensure the following data is accessible:

MassIVE repository

- MSV000096001: We have verified the all raw files have been uploaded, as well as search software (Spectronaut or Spectromine) output files (.sne or .psar, respectively), exported tables in .csv format, and normalized values used in our analyses. This includes the following datasets: SEC-MX from HEK293 used to compare to SEC-DIA, SEC-DIA from HEK293 cells, gSEC and phSEC (HEK293 and HCT116 together), unfractionated global acquired with DIA (non-enriched, both for HEK293 and HCT116 separately), unfractionated enriched phosphopeptides acquired with DIA (HEK293 and HCT116 separately).
- MSV000093915: The datasets of SEC-DIA from HEK293 cells before/after RNase digestion.

Supplementary data files provided with the manuscript: We provide the smoothed and replicate-averaged gSEC and phSEC values, as well as the intensity values from gUF and phUF per protein/phosphopeptide as a resource, in order for the readers to be able to download and explore their proteins of interest.

Additional resources: To facilitate accessibility for readers, we have also uploaded processed versions of the SEC-MX global and phosphorylation data for both HEK293 and HCT116 to an AWS download (https://sec-mx-example-data.s3.amazonaws.com/data_inputs.zip), including all necessary inputs for analysis. This data can also be accessed through the GitHub repository, which is shared with readers (<https://github.com/mjlab-Columbia/SPADE-PTM>).

We hope this additional clarification and data access will address the reviewer's concern and provide readers with the necessary resources for further analysis.

2. What is the relationship between https://github.com/mjlab-Columbia/peak_alignment and SECMX (Size Exclusion Chromatography fractions MultipleXed)? Is pyPA developed for SECMX? There are only few simple scripts. The authors should provide all the data and codes used in this study in GitHub as a resource.

Thank you for the constructive feedback regarding the relationship between our previous GitHub repository (previously https://github.com/mjlab-Columbia/peak_alignment) and the SEC-MX (Size Exclusion Chromatography fractions MultipleXed) analysis. We apologize for the initial lack of organization and clarity in the repository. In response, we have updated the GitHub with a detailed README to enhance user experience, walking readers through the usage of the analysis platform employed in the study.

The primary goal of the GitHub is to provide insight into how we leveraged peak picking and alignment to transition SEC-MS analysis from an interaction-centric to a protein-centric approach, focusing on comparing assembly states across different samples. To support this, we included only the code and inputs necessary for this core aspect of the analysis. Specifically, the repository now demonstrates (a) preprocessing of HEK/HCT data, (b) peak picking, (c) peak alignment, and (d) fold-change calculations, all of which contributed to figures 3-6 in the paper. We did not include scripts for the creation of plots, as these are outside the scope of the repository's intended purpose.

Additionally, we believe that the unclear naming of the software – `peak_alignment` and `pyPA` did not make it clear that the pipeline was created to deal with the data from SEC-MX. We have renamed the GitHub to SPADE-PTM (Size exclusion chromatography Protein Assembly Dynamic Evaluator for Post Translational Modifications) in order to more accurately represent that the pipeline is aimed at this paper. The new link can be found at <https://github.com/mjlab-Columbia/SPADE-PTM>.

We hope that these updates address the reviewer's concerns and provide a clearer resource for readers, helping them better understand and apply the methods used in this study.

3. In Figure 1 – SEC-MX performs comparably to SEC-DIA in coverage and resolution, F, what was the y-axis?

The y-axis should have been labeled “density”, we thank the reviewers for bringing this error to our attention. To clarify, this density is derived from the Kernel density estimation of the Pearson correlations between a protein's elution profiles from both the DIA and SEC-MX datasets. Importantly, the y-axis in this context provides a probability distribution in the form of density; each point on the plot represents the density rather than the probability. To obtain the probability, one would need to integrate the curve to determine the area under it within a specified range, ensuring that the total integral equals 1. We clarified these points also in the figure legend for Figure 1. For further details on this method, we encourage readers to refer to the documentation available at <https://seaborn.pydata.org/generated/seaborn.kdeplot.html>

4. In Figure 1 H and Extended Data Figure 4 – gSEC and phSEC datasets comparisons (A) Venn diagram, what was the background and what was the overlap p value?

We apologize for this missing information. To correct this, we added an appropriate statistical test to the relevant figures and their legends, where there is a Venn diagram depicting overlaps between datasets (Figure 1H - overlapping interactions between SEC-MX and SEC-DIA using SECAT, Extended Data Figure 2F - overlapping interactions between SEC-MX and SEC-DIA using EPIC). These p-values were obtained using a hypergeometric test (probability mass function). Specifically, we assessed the significance of the overlapping interactions between the

two datasets against the background of all identified interactions from MX and DIA, as well as the total possible interaction space based on pairwise combinations of the 4,749 proteins measured in both datasets. In both analyses, we found the p-value to be highly significant, below 0.001. We hope that this clarification enhances the readers' understanding of our analysis and adequately addresses the reviewer's concerns.

5. In Figure 2 – SEC-MX enables phosphopeptide enrichment C, the four figures were very similar? Could the authors highlight the different parts, or evaluate it quantitatively?

We appreciate the reviewer's feedback regarding Figure 2. Indeed, the message we wished to convey through showing the heatmaps (now in the revised Figure 2D) was to highlight the similarity in SEC elution profiles between gSEC and phSEC. Since the samples are fractionated before tryptic digestion and IMAC enrichment, we expect phosphopeptides to have similar SEC elution pattern to that of their parent protein. We believe this similarity indicates the successful enrichment of phosphopeptides from the global sample. To clarify this further, in accordance with the reviewer's suggestion, we have quantitatively evaluated the similarities in elution profiles by calculating the Pearson correlation coefficient distribution between the elution patterns in gSEC and phSEC, for peptides measured in both datasets (peptides were matched based on stripped sequence, Figure 2C). We also updated the legend to specify that approximately 70% of peptides (456 / 627 and 343 / 522, in replicate 1 and 2, respectively) show a correlation greater than 0.5, providing a clearer measure of the similarity. While we have chosen not to highlight specific regions in the heatmaps that are different (now Figure 2D), we dedicate Figure 4 and Extended Data Figure 4 to discuss cases of observed differences in gSEC and phSEC elution patterns. In these subsequent sections we elaborate on specific examples that showcase our interpretation of such differences. For example, in the case of MYH9 that seems to be phosphorylated in its complexed, but not its monomeric form (Figure 4C). We hope these additions support the point of figure 2, which is that the overall agreement between gSEC and phSEC is a testament to the quality of the data. As a side-note, the previous Figure 2 included 4 heatmaps, as we presented a heatmap for each dataset (gSEC, phSEC) separated by replicate. Additionally, we decided to average the replicates in order to reduce measurement noise, and therefore only 2 heatmaps are presented in the main figure (the replicate separated heatmaps are presented in Extended Data Figure 3D alongside the equivalent heatmaps from HCT116 data).

Between clarifications and modifications of the heatmaps and an emphasis on the elution profile correlations, we hope Figure 2 is more accessible to readers.

6. In Figure 3 – A novel protein-centric analysis framework to study assembly-states, what was condition 1 and condition 2? The authors should add the condition names rather than numbers?

We apologize for any confusion caused by our use of "condition 1" and "condition 2" in the figure. These conditions were meant as hypothetical examples and do not represent specific data from the study. To clarify this and avoid further misunderstanding, we have now updated the figure (Figure 4A) to label the conditions as "condition X" and "condition Y". We hope that this revision addresses the reviewer's concern and clarifies the information for readers.

7. In Extended Data Figure 3 – SEC-MX performs on par with published methods, at an order-of magnitude reduction in LC-MS/MS runs, the number of high-confidence interactions for SECMX was very low. Can these methods be ensembled to increase both the number high-confidence interactions and the number of PPIs per MS run?

We thank the reviewers for highlighting this issue with the literature comparison figure (previous Extended Data Figure 3). However, we opted to remove this figure from the current version of the manuscript. We did so for two main reasons. First, we wanted to draw more attention to what we see as the primary strengths of SEC-MX – the capacity to enable PTM enrichment downstream from SEC as well as precise quantitative comparisons between assembly and phosphorylation states, rather than the increase in throughput. Not only did we remove this figure, we also modified the part of the discussion that addresses this point. While we do mention the reduction in instrument time (now elaborated in Figure 1C in “# of MS runs” and “Instrument time (min)”, we understand that TMT multiplexing requires costly reagents, time and expertise, and added this into the discussion (Page 20) so that the readers are able to take all these aspects into consideration in their experimental design.

The second reason for which we decided to remove the comparative plot was that our current analysis is focused on the protein-centric approach for differential analyses as the main use-case for SEC-MX, and not on interaction profiling. In the current version of the manuscript, we use PPI profiling as a means to compare SEC-MX data quality to SEC-DIA, and not as a major application of the method. We were concerned that including this figure might steer the focus to SEC-MX as a tool to profile PPIs and distract the readers from the protein-centric approach.

Nevertheless, we would like to point out that the low number of PPIs presented in the previous plot is associated with the use of interaction profiling tools that are geared towards detection of previously-reported interactions (as opposed to tools that account for all pairwise interactions including “novel” ones), and not due to the nature of SEC-MX. We presented 3 such datasets that were analyzed with “SECAT” (in diamond symbols), 2 of them were based on SEC-DIA and had a similarly low number of PPIs. In contrast, the SEC-MX dataset analyzed with “EPIC” (represented by an orange circle), a program also predicting novel PPIs, yielded a similar number of PPIs as did other co-fractionation studies.

Finally, we agree with the reviewer that it would be very interesting to attempt to enhance the performance of these methods. While it is possible to integrate datasets to increase the number of high-confidence interactions and PPIs per run, this would require significant adjustments to the methodology and even developing new analytical approaches and falls outside the scope of this study. However, this is a promising possibility for future research.

8. In Figure 6 – a peak-centric analysis uncovers different regulation of distinct assembly-states and Extended Data Figure 6 – HEK-HCT Differential and Unfractionated Comparison, E and F, there were only “-” and up of Global? Were there down functions?

We appreciate the reviewer’s interest in the different regions of the scatter plot (now referred to as response groups) that we uncovered based on the scatterplot comparing gSEC and phSEC differential HCT116/HEK293 ratios (Figure 7A). While we previously focused on selected groups, we fully acknowledge that this can be regarded like an incomplete analysis. To correct this, we

now include details about all 9 groups that were identified (# of proteins with peaks in each group) in Figure 7B. Furthermore, we now included all groups in the enrichment analysis shown in Extended Data Figure 7A-B, except for three: the group that shows no significant change in either dimension (group E with 781 proteins), and the two groups with opposite effects (gHEK/phHCT, and gHCT/phSEC) which had 2 members each (not sufficient for enrichment analyses). Specifically, CKB and HUWE1 were found in gHEK/phHCT (response group A), while NOB1 and WAPL were identified in gHCT/phHEK (response group I). The resulting Extended Data Figure 7A-B now includes two additional response groups that were not previously covered in the enrichment analyses. These novel findings indicate that 30-40 proteins exhibited phosphorylated assembly-states with differential abundance in gSEC, without a corresponding differential phosphorylation (by phSEC). We include detailed lists of the proteins in each of the response groups in the supplementary materials for the readers to browse. We thank the reviewer for this comment, as we find that the current analysis portrays a more complete perspective of the response groups.

Reviewer #2 (Remarks on code availability):

What is the relationship between https://github.com/mjlab-Columbia/peak_alignment and SECMX (Size Exclusion Chromatography fractions MultipleXed)? Is pyPA developed for SECMX? There are only few simple scripts. The authors should provide all the data and codes used in this study in GitHub as a resource.

As detailed above, we apologize for the initial lack of organization and clarity in the repository. In response, we have updated the GitHub with a detailed README to enhance user experience, walking readers through the usage of the analysis platform employed in the study.

The primary goal of the GitHub is to provide insight into how we leveraged peak picking and alignment to transition SEC-MS analysis from an interaction-centric to a protein-centric approach, focusing on comparing assembly states across different samples. To support this, we included only the code and inputs necessary for this core aspect of the analysis. Specifically, the repository now demonstrates (a) preprocessing of HEK/HCT data, (b) peak picking, (c) peak alignment, and (d) fold-change calculations, all of which contributed to figures 3-6 in the paper. We did not include scripts for the creation of plots, as these are outside the scope of the repository's intended purpose.

Additionally, we believe that the unclear naming of the software – peak_alignment and pyPA did not make it clear that the pipeline was created to deal with the data from SEC-MX. We have renamed the GitHub to more accurately represent that it is aimed at this paper. It is now named SPADE-PTM (Size exclusion chromatography Protein Assembly Dynamic Evaluator for Post Translational Modifications). The new link can be found at <https://github.com/mjlab-Columbia/SPADE-PTM>.

We hope that these updates address the reviewer's concerns and provide a clearer resource for readers, helping them better understand and apply the methods used in this study.

Reviewer #3 (Remarks to the Author):

Summary:

The authors present a novel approach using Size Exclusion Chromatography (SEC) coupled with Tandem Mass Tagging (TMT) and mass spectrometry to analyze protein-protein interactions (PPIs) and post-translational modifications (PTMs) in HEK293 and HCT116 cells. The integration of SEC with TMT labeling enables high-throughput, quantitative analysis, identifying numerous PPIs and providing insights into protein assembly states and their regulation by PTMs. The authors combined mass spec data with a SECAT analysis that effectively predicts and validates PPIs, constructing high-confidence interaction networks and finally performed an EPIC analysis, that adds robustness by integrating functional association data and complement the SECAT analysis. Additionally, the SEC-MX method was compared to a DIA workflow without labeling and multiplexing. The mass spec measurements have been carried out on a Q Exactive HF instrument.

The study's findings are significant for advancing the understanding of protein interactions and regulation of complex assemblies through PTMs (here phosphorylation) and allows for comprehensive mapping of both together, providing a dual perspective that is often challenging to achieve. The insights gained from the study could advance the understanding of protein assembly states and their regulation, potentially informing research in cellular mechanisms and disease states.

General Remarks:

The data obtained from SEC-MS is technically sound and presented in sufficient detail. However, the manuscript would benefit from addressing issues regarding ratio compression in TMT quantification as well as normalization of phosphorylation sites to a reference proteome, providing more details on the machine learning classifier's validation for SECAT, and discussing false discovery rates in peptide and protein identifications. Additionally, the authors state that the new method saves MS measurement time due to the multiplexing approach and can hence result in more protein-protein interactions per MS run but it is not discussed that the sample preparation procedure of fractionation plus TMT is very time-consuming and the complex experimental design is challenging to establish especially for labs that do not commonly perform TMT experiments. Please find a detailed point-to-point list below:

We thank the reviewer for their careful reading and for pointing out these issues. We addressed each of the points to the best of our ability, as elaborated with more detail below, separately for each point:

- We calculated the TMT ratio compression based on comparing SEC to UF data (Extended Data Figure 6B) and discuss this issue in the Materials and Methods section (Page 24), as well as the Discussion (Page 20).
- We detail below our approach to normalization of phosphopeptide data to a reference proteome.
- We added more details about SECAT performance (Extended Data Figure 2A-B).

- We detail all FDRs used, as well as other search software parameters in the Materials and Methods (Page 25-26).
- We specifically addressed the time estimates of the fractionation procedure as well as TMT cost and experimental design in the Discussion (Page 20), in order to provide full clarity on these aspects of the method.

Minor Points:

- The manuscript would highly benefit from an overview figure that describes what kind of samples have been processed with what kind of workflow and analyzed with what kind of software pipeline and additional software packages. It is partially done in Figure 1 and 2 but only until MS measurements without data analysis pipeline used afterwards. A full overview of all experiments, pipelines and so on would help the reader to get an easy entrance point

Thank you for the thoughtful suggestion to provide a comprehensive summary of the samples, workflows, and software pipelines used in our study. We fully agree that this would increase the clarity about the experimental design for the readers. Therefore, we have revised the manuscript to include detailed overviews of the pipeline for each experiment, extending beyond the MS measurement step. Specifically, for the comparison of SEC-MX to SEC-DIA we added which cell-lines were used (HEK293), the number of replicates (4, divided to 2 for MX and 2 for DIA), and the analysis steps: “Signal processing” and “protein-protein interaction analysis” (Figure 1A).

For the experiment generating the gSEC and phSEC datasets from HEK293 or HCT116 cell-lines, we added the cell-lines used, number of replicates, as well as “Signal processing” and “Peak analysis” (Figure 2A) so that it is clear that this is distinct from the PPI analysis conducted in the first section. However, we decided to keep the details of the specific software packages for PPI analysis, including SECAT and EPIC, as well as the details about the Peak Analysis software, in the Materials and Methods section (Page 28-30), where we felt these technical details were best suited. Additionally, the provided GitHub repository (SPADE-PTM: <https://github.com/mjlab-Columbia/SPADE-PTM>) offers further details about the Peak Analysis packages and the steps involved.

Finally, in Figure 5B, we provide an additional overview focusing on the multiplexing protocol for the HEK293 and HCT116 cell lines. This figure aims to clarify the workflow for this step, providing readers with a clearer understanding of how these two cell lines were multiplexed.

We hope these revisions offer a clearer entry point for readers and adequately address the reviewer’s concern.

Figure 1: Maybe this is just due to zooming in but why does the profile comparison between DIA and MX in (E) look to different compared to the full profiles in D?

Indeed, both heatmaps (Figures 1D and 1E) represent the same dataset. The observed difference is, as suggested by the reviewer, primarily due to the zoomed-in view in Figure 1E and a slight misalignment between the axes in the two figures. To clarify the differences in scale, we have added the number of proteins represented in each heatmap. In addition, we note in the figure

legend describing panel 1E that this is a subset from figure 1D. We hope these adjustments will provide greater clarity for the readers and address the reviewer's concerns.

Figure 1: (F) it is unclear what is plotted on the y-axis

We apologize for our oversight and thank the reviewer for catching it. We have now labeled the y-axis as "density," which reflects the Kernel density estimation of Pearson correlations between a protein's elution profiles from both the DIA and SEC-MX datasets. This density represents a probability distribution, with each point indicating density rather than probability. To obtain probability, the curve must be integrated over a specified range, ensuring the total area equals 1. We have pointed this out in the legend. For more details, we refer readers to the Seaborn KDE Plot documentation (<https://seaborn.pydata.org/generated/seaborn.kdeplot.html>). We hope this clarifies our method for evaluating the similarities in SEC elution between DIA and MX and addresses the reviewer's concerns.

Figure 1: (G) would be good to clarify that this numbers are from the SECAT analysis directly in the figure (headers maybe)

Thank you for the suggestion. We have done as suggested and added the title 'SECAT network parameters' to the header of the table in Figure 1G. Hopefully this makes the table clearer for readers.

Figure 2: (A) it is not clear from the figure whether the IMAC enrichment was performed on the total TMT pool or on the TMT pool of cells but for each of the 54 fractions separate. This information is also missing in the method section.

Thank you for raising this important point. To clarify, the IMAC enrichment was performed on each of the pooled 18 channels TMT mixes. Each such mix included 18 multiplexed fractions: 9 from HEK293 and 9 from HCT116 (rather than on the 54 separate fractions), as elaborated in Extended Data Figure 1B. In fact, multiplexing 18 fractions together was necessary to provide sufficient input for the IMAC enrichment, and this approach is one of the key strengths of the SEC-MX method. In response to your question, we have added a figure (Figure 5B) to illustrate the mixing scheme in more detail, and put it with regards to the IMAC enrichment. We also added an explicit description of how the mixes were pooled together for IMAC enrichment in the Materials and Methods section (Page 24). We hope these additional details will make this process clearer to the readers.

Extended Data Figure 3:

- The reference data sets for performance comparison are quite outdated, for example the crosslinking data set is from 2021, Bartolec et al. (2023) (<https://doi.org/10.1073/pnas.2219418120>) reported 28,910 unique residue pairs, 4,084 unique proteins, and 2,110 unique protein-protein interactions from a

crosslinking experiment coupled with SEC. (approximately 482 interactions per MS run)

Thank you for bringing this important reference to our attention. Following your comments (including the next one) and other reviewer's feedback, we reconsidered this figure and decided to remove it from the revised version of the manuscript. Mainly, we wanted to put more emphasis on SEC-MX advantages in enabling PTM enrichment as well as precise quantitative comparisons between assembly and phosphorylation states, and less emphasis on the throughput aspect of the method. Moreover, we were concerned that a full figure dedicated to PPI identification will steer the reader's attention away from the focus on the protein-centric peak analysis of SEC-MX data. Finally, as pointed out below, we agree that the PPI per MS-run calculation might have been too simplistic and did not fully and accurately reflect the method comparison.

However, your feedback also highlighted the need to accurately represent the advancements in crosslinking experiments. We have now incorporated the Bartolec et al. (2023) reference into the manuscript, specifically in relation to recent works studying the protein-interactions network in HEK293 cells (Page 11, reference 42), and we appreciate you bringing this relevant study to our attention. We hope these changes clarify our approach and address the concerns raised, ultimately helping readers better understand the contributions of our work.

Extended Data Figure 3: Taking interactions per MS run can be quite misleading because it does not consider the MS run time. One could choose longer gradients to alter the position of the study in the diagram. In fact, the total run time of SEC-DIA in this manuscript is 70min, while DIA (unfractionated) gradients are 160min and SEC-MX gradients are 150min. Was the total run time somehow normalized before plotting it? It is not stated in the figure legend

We acknowledge that the metric "PPI per MS run" in the previous Extended Data Figure 3, which did not normalize for instrument time, could present an unfair comparison. We previously chose this metric since obtaining accurate instrument time information for every study referenced has proven to be challenging (often this information is not reported). For this reasons, and other arguments related to the current focus of the paper (elaborated above), we chose to remove this figure from the revised version of the manuscript.

To address the concern of reporting instrument time more clearly, we have added the total instrument time for our SEC-DIA and SEC-MX experiments to the comparative table in Figure 1C. This table now highlights the number of MS-runs as well as the time differences between the two methods. Specifically, we found that over the course of 57 DIA runs, the total instrument time was 3,990 minutes (with a 70-minute gradient), while 8 SEC-MX mixes required approximately 1,200 minutes of MS time (with 150-minute gradients). We hope these revisions clarify the time-related aspects of our methods and address the concerns raised regarding run time normalization.

Extended Data Figure 4:

(A) Is the peptide intersection of HEK and HCT cells together or just one cell type?

(B) If the overlap between gSEC and phSEC is so little how can the profiles between both methods be so similar? If just these 960 overlapped peptides are plotted here, then this must be stated somewhere, at least in the figure legend

Thank you for pointing out that this was not clearly explained. The previous Extended Data Figure 4A (now Extended Data Figure 3B) presented a Venn diagram of the peptide identifications between the gSEC and phSEC datasets, where each dataset included the union of peptide identifications of HEK293 and HCT116 cells. Notably, since the overlap of peptide IDs between HEK293 and HCT116 cells is nearly complete (a result of the mixing scheme used), the gSEC-phSEC peptide identifications overlap based on HEK293 and HCT116 individually is nearly the same.

In addition, we would like to note that the corresponding heatmaps in Figure 2D (previously in Figure 2C) had changed: while we previously presented 4 different heatmaps, 2 for gSEC and 2 for phSEC - separated by replicate, we now opted to present the heatmaps based on the replicate-averaged signal. This was following our decision to conduct the downstream analyses on the replicate-averaged signal, which reduces measurement noise for a more robust peak-calling. As a result, the heatmaps cover only 909 peptides, (960 overlapping between gSEC and phSEC as presented in the Venn diagram, and reduced to 909 due to the requirement that all peptides would have been identified in both cell-lines).

To complement this, the corresponding Extended Data Figure (Extended Data Figure 3D) now contains the heatmaps that are separated by replicate: 4 for each cell-line (HEK293 - top, HCT116 - bottom), showing both gSEC and phSEC in 2 replicates each. For the sake of comparing between these heatmaps, we present only the 286 peptides (based on stripped sequence) that were consistently measured in all 8 datasets. Notably, the main reduction in the number of peptide overlap stems from differences between the global and phospho-enriched datasets (as shown in Extended Data Figure 3B), rather than the differences between HEK293 and HCT116 cells, with further limitations arising from replicate variation.

In response to this comment, we now indicate the number of peptides represented in each heatmap on its y-axis and detail in the legend what is the specific group of peptides that is plotted. Additionally, we include Table 1 in the main text - with protein and peptide identification numbers based on the replicate-averaged signal, and Extended Table 1 - with protein and peptide identification numbers broken down by replicate.

Method section:

- **The FDR settings used for searches must be mentioned in the data analysis section, but this is not the case, please add this information.**
- **Although it is stated that the BSG factory settings are used it is good practice to write at least some details about search settings that have been used for example: modifications used, ppm error on MS1 or MS2, was the quantification performed on MS1 or MS2 level? And so on. Spectronaut 16 is quite old now and factory settings certainly have changed.**

Thank you for highlighting this aspect, we agree these important details should have been included in the Materials and Methods section and apologize for this oversight. We have modified the “Searches” section under “Data Analysis” in the Materials and Methods section (Page 25-26). It now reads as follows:

“For SEC-MX, search parameters were set to BGS factory settings for TMTpro 18 channels, modified without automatic cross-run normalization or imputation. These parameters were as follows: false discovery rate (FDR) of 0.01 for identification of peptide-spectral matches, peptides and protein groups. Peptide maximum charge of 4, maximum length of 35. Quantity MS level – MS2 (base quantity unit was TMT reporter intensities), Minor group (peptide) was quantified by summing using the top 3 method. Major group (protein group) was quantified by summing the top 5 peptides, MaxLFQ option was “on”. For the phosphoproteomics dataset (phSEC), raw files were searched similarly with an additional variable phospho(STY) modification, with PTM localization workflow as follows: phosphosite flanking region – 7 amino acids, multiplicity “True”, PTM consolidation method – “SUM”, phosphosite localization probability filter set to 0.

For DIA runs we used the equivalent parameters in SpectroNaut, with minor modifications: Major group (protein group) was quantified by summing the top 3 peptides, Quantity MS level – MS2, Quantity type – Area, for phUF data an additional variable phospho(STY) modification was defined, PTM localization workflow as follows: phosphosite flanking region – 7 amino acids, multiplicity “True”, PTM consolidation method – “SUM”, phosphosite localization probability filter set to 0.5. Cross run median normalization and global imputation were used for global expression analysis (HEK-HCT unfractionated samples). Peptide spectral matches (PSMs), peptides and protein group data were exported for subsequent analysis, as well as a PTM site report when applicable.”

We appreciate your suggestion to include these important details and hope this enhances the clarity for readers.

Method section: Conversion of empty or NA values to zero is one methods that can be used but also not the best for this kind of analysis. Have been other techniques evaluated for example Nearest neighbor imputation?

We appreciate the reviewer's concern regarding the conversion of empty or NA values to zero. We have tested several methods to deal with missing values in SEC data, including the use of zeros, NA values, and various imputation techniques, to evaluate their impact on our analysis. Our testing indicated that using zeros performed best in terms of identifying interactions and managing the dataset, with no detrimental effects on the peak picking and alignment stages of our analysis. Importantly, the peak picking algorithm is unaffected by whether zeros or imputations are used, as it incorporates minimum intensity filters that exclude the full elution profile below certain thresholds (1000, higher than imputed noise).

Additionally, converting to zeroes is in-line with the meta-analysis by Skinnider & Foster (2021), which demonstrates that using zeros outperformed both NA values and imputed noise in terms of area under the curve (AUC) for interaction modeling in co-fractionation assays (see their Figure 2). We have added this information to the Materials and Methods section (Pages 26-27),

in order to provide better clarity for the readers. We hope that this clarifies the rationale behind our choices and addresses the reviewer's concerns effectively.

Skinnider, M. A. & Foster, L. J. Meta-analysis defines principles for the design and analysis of co-fractionation mass spectrometry experiments. Nat. Methods 18, 806–815 (2021).

Method section: Please provide detailed information on the training and validation of the machine learning classifier used in SECAT analysis. Include performance metrics such as precision, recall, and F1 scores.

Thank you for your interest in the performance of the machine learning classifier used in our SECAT analysis. To address these concerns, we have added a plot that illustrates SECAT's training and classification performance on both the DIA and MX datasets (Extended Data Figure 2A). These details were directly derived from the outputs of the SECAT pipeline, which we believe provides a clear overview of its performance metrics. Specifically, the S-value represents the True Positive Rate, or recall (y-axis), while the q-value reflects a measure similar to precision, where a lower q-value generally indicates higher precision (x-axis). Note that when reporting interaction, we use a q-value cutoff between 0.05 and 0.1 (Extended Data Figure 2C). This is in addition to the previously included d-score density distributions (Extended Data Figure 2B), showing a clear distinction between decoy and true targets in both SEC-MX and SEC-DIA. We encourage the reviewer and readers to refer to SECAT for a comprehensive evaluation of its ability to train and validate its models (Rosenberger et al., 2020). We hope this response clarifies the training and validation aspects of SECAT and effectively addresses your concerns.

Rosenberger, G. et al. SECAT: Quantifying Protein Complex Dynamics across Cell States by Network-Centric Analysis of SEC-SWATH-MS Profiles. Cell Syst. 11, 589-607.e8 (2020).

Method section: As far as I get it 2.5 ug peptides were injected for each MS run. I think this is quite a lot, especially for 70 min gradient run. Was the carry over evaluated between the fractions?

Thank you for bringing this issue into our attention. We routinely evaluate the concentration of any sample before injecting for LC-MS/MS, by nanodrop quantification against a blank buffer (A205, using the Scopes method). We consider that the resulting concentration estimates are trending towards over-estimation, and we monitor our Total Ion Chromatogram (TIC) plots to guarantee that the signal is in the expected range. To satisfy the reviewer's concerns, we tested sample carry-over by measuring 3 consecutive "empty" runs (injecting solvent only) after a 2.5ug injection using the same instrument method, in triplicate. The results are attached below, and indicate that carry-over between samples is minimal.

Major Points:

- Quantitation of TMT experiments can be tricky, the issue of ratio compression, commonly observed in TMT-based quantification, is not thoroughly addressed in the method section. Ratio compression can lead to inaccurate quantification of peptide abundances, affecting the interpretation of interaction strengths and protein assembly states.

We appreciate the reviewer's constructive feedback regarding the quantitation challenges associated with TMT experiments, specifically concerning the issue of ratio compression. In response, we calculated the compression of our DDA TMT mixes compared to unfractionated DIA runs by summing the SEC intensities per protein and calculating the ratios between the HCT116 and HEK293 samples. Our analysis revealed that the TMT data was compressed by a factor of 0.69, as shown in Extended Data Figure 6B (1/1.45). This information is detailed in the Materials and Methods section of the paper (Page 24). Moreover, we included a discussion on TMT ratio compression in the text (Page 20) in the discussion section detailing the caveats associated with conducting TMT labeling.

We acknowledge that this compression could influence the log₂-fold ratios presented in the quantification of assembly state enrichment. However, we believe that this issue is not detrimental in SEC-MX for several reasons. First, our multiplexing scheme somewhat minimizes the issue of data compression since all comparisons are conducted within the same TMT mix, where TMT actually tends to be highly precise and therefore allows us to call significant

differences at smaller fold changes. Additionally, taking a protein-centric perspective, we do not compare differential ratios between different proteins, but rather between assembly-states of the same protein. While the compression may affect the log₂ ratios and their cutoffs, we ensure that our findings are not biased by this compression by refraining from using cutoffs on the unfractionated data during our analysis. In addition, we tried to minimize direct comparisons of fold-changes between gSEC and phSEC, which might not be affected by ratio compression to the same degree, given their different complexities (Savitski et al., 2013; Rauniyar and Yates, 2014).

We hope these clarification addresses the reviewer's concerns and enhances the understanding of our methodology for the readers.

Savitski, Mikhail M., et al. "Measuring and managing ratio compression for accurate iTRAQ/TMT quantification." Journal of proteome research 12.8 (2013): 3586-3598.

Rauniyar, Navin, and John R. Yates III. "Isobaric labeling-based relative quantification in shotgun proteomics." Journal of proteome research 13.12 (2014): 5293-5309.

- Additionally, the accuracy of the TMT quantitation can be enhanced by normalization of the phospho-peptide intensities (areas) to a reference proteome that was measured before pooling the fractions. Maybe this was already done, and I have overlook it but it would be good to have a detailed description in the method section. If this was not performed, there would be a bias in the conclusions about assembly state regulation by phosphorylation towards protein abundance differences instead of phosphorylation differences

We appreciate the reviewer's insightful suggestion regarding the normalization of phosphopeptide quantifications. However, we intentionally chose not to normalize the TMT quantification of phosphopeptide intensities to a reference proteome. We believe that both scenarios—where phosphorylation changes align with assembly-state abundance and where they do not—carry important biological significance. To preserve this valuable information, we explore both dimensions of the data in our manuscript.

By separating the two dimensions (abundance and phosphorylation levels, Figure 7A-B) we highlight studying each one independently, as well as investigating their concordances and discrepancies. For example, we define 9 "response" groups that differ based on their 3 possible "behaviors" in each dataset (up in HEK, up in HCT or not significant – NS). We identified groups of proteins with assembly-states that differ on the phosphorylation level, without concomitant difference in abundance (groups B: gNS|phHCT, group H: gNS|phHEK) and vice versa (group D: gHEK|phNS, group F: gHCT|phNS). We also hypothesized that these groups might be regulated by different kinases and conducted an analysis of kinase-target enrichment to support this (Extended Data Figure 7B). In other words, through our direct comparison between gSEC and phSEC, readers are able to immediately see if a phosphorylation change is due to an abundance change as they would see that both values show strong concordance in the scatter plot.

Our approach focuses on identifying these differences independently as they could potentially be regulated by distinct mechanisms. We do not compare phosphorylation abundances to global abundances directly; instead, abundances are only compared within gSEC and within phSEC. Our comparative analysis of gSEC and phSEC aims to ascertain whether regulation

occurs at the abundance level or the phosphorylation level, and normalizing phSEC to the global data would obscure these differences. We hope that this response clarifies our methodology and addresses the reviewer's concerns.

- What kind of control was used for protein SEC to test whether complexes are unaffected by the procedure itself? O'Reilly et al 2023 (<https://doi.org/10.15252/msb.202311544>) for example could show that some protein complexes dissociate during the fractionation process but can be stabilized using crosslinking reagents. If this is also the case for protein complex presented in this study the conclusion about difference between phosphorylation states of monomers and full complexes would be biased.

Thank you for your insightful comments regarding the potential limitations of traditional SEC-MS methods, particularly concerning the stability of protein complexes during fractionation. We acknowledge this as a valid point and appreciate your reference to the work by O'Reilly et al. (2023), which highlights the issue of complex dissociation. Indeed, SEC tends to favor more stable complexes over transient ones, due to the long sample processing time for fractionation and the high volumes used in fractionation. While we acknowledge that this is an important aspect to be considered in choosing SEC and other co-fractionation methods to analyze protein-interactions, this issue is relevant for co-fractionation in general, and not specific to SEC-MX. Therefore, we feel that it is out of the scope of the current study, which is focused on developing and evaluating the compatibility of labeling and multiplexing in SEC-MS.

Moreover, due to the bias in SEC towards identifying stable interactions, we believe that our estimates for the percentage of proteins that elute exclusively in their assembled form (Figure 3C-D) are lower bound estimates, for both our global and phosphorylation data. Despite this being the case, we observed that most proteins eluted within the complex region, with a notably lower percentage found in the monomer region (see Figure 3C and Extended Data Figure 3G). If the majority of protein complexes would have dissociated, we would expect to see a much greater number of proteins in the monomer region. This trend is even more pronounced in our phosphopeptide data (Figure 3D and Extended Data Figure 3G), indicating that, while we cannot exclude the possibility of dissociation during the experimental procedure, the data suggest that most complexes likely remain intact during the analysis.

Taken together, while we agree that SEC-MX might benefit from the incorporation of crosslinking techniques, we believe this is outside the scope of the current manuscript, which aims to demonstrate the multiplexing capability for phosphopeptide enrichment, how to manage such data, and the value it provides. Nonetheless, we consider it an interesting future direction to explore the incorporation of crosslinking into SEC-MX added it as a future perspective. We added this to the discussion section along with describing the challenge of capturing transient interaction and pointed out these limitations in the revised manuscript (Page 20).

We hope that this clarification addresses the reviewer's concerns and that the modification we introduced in the discussion would lead to a better portrayal of the method's limitations.

Supplementary data:

- As supplementary data a list of protein names (gene names) was provided but there is no title nor legend or description available. I think this should be added directly in the document

We apologize for this misunderstanding; we had intended to include a detailed legend in the first sheet of the supplementary file. We edited the supplementary materials and verified they contain a detailed explanation of the provided data.

- Data availability: I could not find the .sne files from Spectronaut. It would be nice to provide the full Spectronaut and SpectroMine analysis files

We apologize that these files were not previously provided. We updated the MassIVE repository to include all search files (“.sne” for spectomine and “.psar” for spectronaut) as appropriate, as well as the exported values in tabular format. Thank you for pointing out this issue.

The updated MassIVE repository can be found here:

accession number: MSV000096001.

DOI: [https://doi.org/ doi:10.25345/C58912314](https://doi.org/doi:10.25345/C58912314)

FTP Download Link: <ftp://MSV000096001@massive.ucsd.edu>

Log in with username: MSV000096001_reviewer, password: secmx1234

Point 1:

Reviewer 1: The authors have successfully addressed all of my comments, with the exception of the third major point.

Redacted

“The authors have successfully addressed all of my comments, with the exception of the third major point. However, I recognize the significant improvements in both the quality and content of the revised manuscript, and I therefore recommend it for publication in its current form. It would be interesting if, in future work, the authors could explore whether they can capture phosphorylation-dependant complex reassemblies following rapid stimulation rather than focusing solely on basal states.”

We thank the reviewer for the positive feedback and their acknowledgment that the introduced changes significantly enhance the manuscript. We were happy to hear that the reviewer agrees that the additional content improves readers' understanding of the method's potential biological relevance and applicability. While this proof-of-principle study did not focus on dynamic or rapid perturbations, we concur that future applications should explore such scenarios, including drug treatments (e.g., Torin 1 or other mTOR inhibitors), stress responses (e.g., sodium arsenite or other acute stressors), or immune stimulation (e.g., lipopolysaccharides or other adjuvants), among many others. We fully intend to investigate such perturbations in future SEC-MX studies. However, we think that they go beyond the current study that introduces the method and its potential. Additionally, we hope this proof-of-principle publication inspires other groups to adopt SEC-MX for studying the interplay between protein assembly states and post-translational modifications in their systems of interest.

Point 2:

Reviewer 1: Unfortunately, the example data is currently unavailable, as the S3 file cannot be downloaded due to access denial at this time.

Since the data is inaccessible, I'm unable to test the code's functionality at this moment. Despite this, the code is well-structured, clearly documented, and highly readable.

P.S: This is the link in the repo I can't access.

https://sec-mx-example-data.s3.amazonaws.com/data_input.zip

https://sec-mx-example-data.s3.amazonaws.com/data_input.zip

Redacted

We apologize for this mistake; the links were not pasted correctly. The correct link is the following:

https://sec-mx-example-data.s3.amazonaws.com/data_inputs.zip

This link has been updated in all its occurrences (in the manuscript file and in the Github repository).

Point 3:

Reviewer #2 (Remarks on code availability):

The code looks correct and for each step, there are explanations. But I do not use macOS and have not actually run the code.

To facilitate the use of SPADE for non-macOS users, we verified that the code runs smoothly and successfully on Windows and updated the README file accordingly to make this clear.

Point 4:

Reviewer 3: I am still concerned about the 69% ratio compression and labelling efficiency of below 95% for two of the TMT mixes but the authors openly reported this issue using figures and discussed it partially in the main text. I appreciate the openness at this point. If the other reviewers are not concerned, I am fine with it like it is because it is mentioned that this is a proof-of-concept study and there is room for improvement.

Redacted

We appreciate the reviewer's comments and acknowledge the challenges associated with TMT labeling, as addressed in the text. To the best of our knowledge, a ratio compression to 69% is within the expected range for MS2-based TMT quantifications and aligns with previous reports cited in the manuscript. While ratio compression could potentially be improved using MS3-based quantifications, this requires an MS3-capable instrument. We would like to add that the increased precision of TMT-labeling based quantifications allows researchers to use lower fold change ratio cutoffs than are normally used for label free quantifications, partially compensating for the ratio compression (PMID 32203386).

Regarding labeling efficiency, we apologize for the confusion. As reported in the "TMT labeling" section of the methods, the labeling is actually 98.3%, which aligns well with accepted standards in the field (PMID: 30967486).

We believe the reviewer's mention of <95% efficiency refers to Supplementary Figure 3A, which reports IMAC (phosphopeptide) enrichment efficiency, which is in line with most standard studies in the field of phosphoproteomics (PMID: 29988108). We apologize for any confusion and have clarified the captions in Supplementary Figure 3A to ensure it is more understandable for readers.